# Variational Routing: A Scalable Bayesian Framework for Calibrated Mixture-of-Experts Transformers

Albus Yizhuo Li [1]    Matthew Wicker [1]

## Abstract

Foundation models are increasingly being deployed in contexts where understanding the uncertainty of their outputs is critical to ensuring responsible deployment. While Bayesian methods offer a principled approach to uncertainty quantification, their computational overhead renders their use impractical for training or inference at foundation model scale. State-of-the-art models achieve parameter counts in the trillions through carefully engineered sparsity including Mixture-of-Experts (MoE) layers. In this work, we demonstrate calibrated uncertainty at scale by introducing Variational Mixture-of-Experts Routing (VMoER), a structured Bayesian approach for modelling uncertainty in MoE layers. VMoER confines Bayesian inference to the expert-selection stage which is typically done by a deterministic routing network. We instantiate VMoER using two inference strategies: amortised variational inference over routing logits and inferring a temperature parameter for stochastic expert selection. Across fine-tuning tested foundation models, VMoER improves routing stability under noise by 38%, reduces calibration error by 94%, and increases out-of-distribution AUROC by 12%, while incurring less than 1% additional FLOPs. These results suggest VMoER offers a scalable path toward robust and uncertainty-aware foundation models.

## 1. Introduction

Foundation models are increasingly deployed in open-world settings characterised by distributional shift, partial observability, and high-stakes decision-making (Bommasani, 2021; Szpruch et al., 2025). In such settings, the ability to quantify uncertainty is central to responsible deployment, governing when predictions should be trusted, deferred, or interrogated (Kendall & Gal, 2017; Babbar et al., 2022; Wicker et al., 2025). However, contemporary foundation models overwhelmingly rely on deterministic training and inference pipelines that obscure epistemic uncertainty and yield overconfident predictions outside their training support (Gal). While Bayesian methods provide a principled framework for uncertainty quantification, their computational and memory costs have rendered them largely incompatible with the scale at which modern foundation models operate (Wenzel et al., 2020). The tension between the necessity of uncertainty-aware reasoning and the engineering realities of trillion-parameter architectures has resulted in a critical gap between theory and practice. Bridging this gap requires uncertainty estimation mechanisms that are both statistically grounded and natively compatible with the sparsity and modularity that enable foundation model scaling.

Mixture-of-Experts (MoE) layers have become the standard for scaling foundation models, replacing dense Feed-Forward Networks (FFN) with a set of specialised "experts" networks to achieve massive parameter counts at constant inference cost (Shazeer et al., 2017). At the time of writing, every non-proprietary foundation model in the top fifty of LMArena's text leaderboard employs an MoE architecture (Chiang et al., 2024).[1] Central to MoEs is the *routing mechanism*, a lightweight network that selects a sparse subset of experts for each token. In standard implementations, this router operates in three stages: (1) a **linear projection** computes similarity scores (logits); (2) a **Softmax** function projects them onto probability simplex; and (3) a hard **Top-K** operation enforces sparsity.

While efficient and scalable, routing networks are prone to various failure modes. During training, MoE routers may experience expert collapse where routers send a disproportionate fraction of tokens to a small subset of experts (Guo et al., 2017); during fine-tuning, selection drift can substantially hinder performance (Zoph et al., 2022), and routing decisions are often sensitive to numerical precision and input perturbations (Fedus et al., 2022; Lepikhin et al., 2020). In Figure 1a we illustrate this

---

[1] Department of Computing, Imperial College London. Correspondence to: Matthew Wicker <m.wicker@imperial.ac.uk>.

*Proceedings of the $43^{rd}$ International Conference on Machine Learning*, Seoul, South Korea. PMLR 306, 2026. Copyright 2026 by the author(s).

---

[1] Detailed discussion in Appendix G.

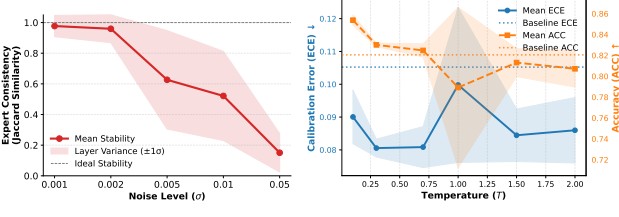

*(a) Brittleness of Deterministic Routing*    *(b) Calibration-Accuracy Trade-off*

*Figure 1.* **Motivation. (a)** Deterministic routing is empirically brittle to input noise. **(b)** Simple low-temperature sampling ($T<1$) creates a "sweet spot" that improves both Calibration (Blue $\downarrow$) and Accuracy (Orange $\uparrow$) over the baseline (dotted). Both motivational experiments' details can be found in Appendix B.

instability by demonstrating the brittleness of routing decisions in response to small random input noise.

One effective approach to mitigating routing failures is the introduction of stochasticity (Fedus et al., 2022) which has been a tried-and-trusted method to stabilise training dynamics, improve robustness, and capture uncertainty. In Figure 1b we show that replacing hard Top-K with low-temperature sampling ($T < 1$) not only improves robustness (as shown in (Fedus et al., 2022)) but significantly reduces calibration error (ECE). In this paper, the critical question we raise is *how* to add stochasticity. We adopt a principled Bayesian perspective adding stochasticity into MoE layers. Prior Bayesian approaches in foundation models focus on weight uncertainty, modelling uncertainty within attention or experts and often incurring substantial inference-time overheads (Fan et al., 2020).

We propose **Variational Routing** for MoE (**VMoER**), a framework that shifts the site of inference from the parameters of attention or experts themselves to the router's decisions. Viewing MoE routing probabilistically, we observe that Top-K routing is inherently multi-label, while existing routers rely on single-label likelihoods (softmax vectors), preventing modelling of expert correlations. Further, a Bayesian perspective allows us to view previous stabilisation strategies (load balancing regularisation (Lepikhin et al., 2020) and auxiliary losses (Zoph et al., 2022)) as implicit priors.

To structure our Bayesian approach, we propose two complementary forms of variational inference for routing networks: (1) *Logit-Space Inference:* we consider applying amortised variational inference directly to the similarity scores to capture expert correlations and (2) *Selection-Space Inference:* which learns a latent, input-dependent temperature to dynamically scale decision boundaries. Both approaches side-step the need for and computational burden of weight-space posteriors.

We conduct a rigorous evaluation across three distinct SOTA architectures—**Granite-MoE** (IBM, 2024), **Qwen-MoE**

(Qwen, 2024), and **DeepSeek-MoE** (Dai et al., 2024). Our results demonstrate that VMoER consistently improves model reliability, increasing routing stability against perturbation by 38%, reducing in-distribution calibration error (ECE) by up to 94%, and unlocking a latent uncertainty signal that improves Out-of-Distribution detection AUROC by 12%. Further, we verify that VMoER incurs negligible activation memory and FLOPs overhead ($< 1\%$). Unlike standard Bayesian LLM approaches that attempt inference in either whole parameter space (Yang et al., 2024; Wang et al., 2024), attention (Chen & Li, 2023) or experts (Dialameh et al., 2025), our targeted intervention on the router offers a computationally tractable pathway for enhancing the trustworthiness of large-scale foundation models. We summarize our contributions:

- We formalize Mixture-of-Experts routing as a *latent variable model*. By shifting inference from high-dimensional weights to the decision manifold, we treat standard heuristics like load balancing as implicit Bayesian priors.
- We introduce two methodological paths for amortized inference: *Logit-Space Inference* to capture expert correlations and *Selection-Space Inference* to learn input-dependent temperatures for dynamic decision boundaries.
- We demonstrate that VMoER increases routing stability by 38% and reduces calibration error by up to 94% across Granite, Qwen, and DeepSeek-MoE architectures, while incurring negligible ($< 1\%$) computational overhead.

**Conflict of Interest Disclosure** The authors declare that they have no competing financial interests or personal relationships that could have appeared to influence the work reported in this paper. Computational resources for this work were provided by the Department of Computing at Imperial College London.

## 2. Probabilistic Mixture-of-Experts Routing

We begin by describing and formalising the standard, deterministic routing network. We consider a Mixture-of-Experts (MoE) layer embedded within a Transformer block. Let $\mathbf{u} \in \mathbb{R}^D$ denote the input token representation and $\{E_i\}_{i=1}^N$ be a set of $N$ expert networks parameterised by weights $\mathbf{W}_{\text{exp}}$. The subject of our focus is the *routing mechanism*, parameterised by learnable weights $\mathbf{W}_r \in \mathbb{R}^{D \times N}$, which determines the sparse assignment of tokens to experts.

**Deterministic Routing** In standard implementations, the router operates as a deterministic directed graph, processing variables in the following sequence (as shown in Figure 2):

| | | |
|---|---|---|
| **Logits:** | $\mathbf{l} = \mathbf{u}\mathbf{W}_r$ | (1) |
| **Probs:** | $\mathbf{p} = \text{Softmax}(\mathbf{l}), \quad \mathbf{p} \in \Delta^{N-1}$ | (2) |
| **Selection:** | $\mathbf{z} = \text{Top-K}(\mathbf{p}), \quad \mathbf{z} \in \{0,1\}^N$ | (3) |

The final layer output $\mathbf{y}$ is the weighted sum of expert out-

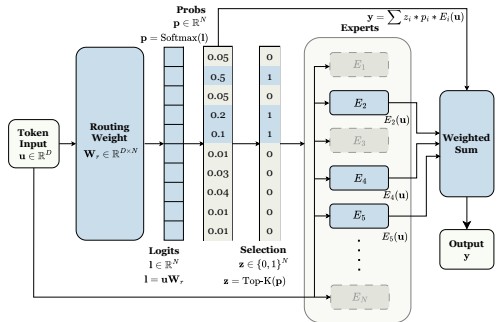

*Figure 2.* **Standard deterministic router architecture.**

puts, gated by both the probability parameters $p_i$ and the discrete selection mask $z_i$: $\mathbf{y} = \sum_{i=1}^{N} z_i \cdot p_i \cdot E_i(\mathbf{u}; \mathbf{W}_{\text{exp}})$

### 2.1. MoE as a Latent Variable Model

To reason about how to conduct inference, we reformulate MoE layer as a probabilistic generative model, visually depicted as a PGM in Figure 3. We postulate that the observed output $\mathbf{y}$ is generated through a latent decision process involving logits $\mathbf{l}$, probability simplex $\mathbf{p}$, and discrete selection mask $\mathbf{z}$.

**Generative Process** We define the generative process for a single token as follows:

(1) **Latent Logit Generation (l):** Given input $\mathbf{u}$ and router weights $\mathbf{W}_r$, the model generates a latent logit vector from the conditional distribution $p(\mathbf{l}|\mathbf{u}, \mathbf{W}_r)$. In deterministic routing, this collapses to a Dirac delta: $\delta(\mathbf{l} - \mathbf{u}\mathbf{W}_r)$.

(2) **Probability Projection (p):** The logits are transformed into the simplex via $\mathbf{p} = \text{Softmax}(\mathbf{l})$. The uncertainty in the logits propagates through the Softmax function, establishing the dependency relation $p(\mathbf{p}|\mathbf{l})$.

(3) **Sparse Expert Selection (z):** The discrete selection mask $\mathbf{z}$ is sampled based on the probabilities $\mathbf{p}$, subject to the sparsity constraint $\|\mathbf{z}\|_0 = k$. We denote this conditional dependency as $p(\mathbf{z}|\mathbf{p})$.

(4) **Observation (y):** Finally, the output is generated by the selected experts, conditioned on the active path defined by $\mathbf{z}$ and $\mathbf{p}$: $\mathbf{y} \sim p(\mathbf{y}|\mathbf{u}, \mathbf{z}, \mathbf{p}, \mathbf{W}_{\text{exp}})$.

**The Bayesian Inference Challenge** Ideally, for a new input $\mathbf{u}^*$, we wish to compute posterior predictive distribution by marginalising over all global parameters $\Theta = \{\mathbf{W}_r, \mathbf{W}_{\text{exp}}\}$

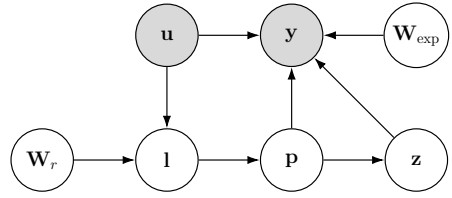

*Figure 3.* **MoE Routing PGM.** Probabilistic graphical model formulation of MoE routing treating decisions as latent variables.

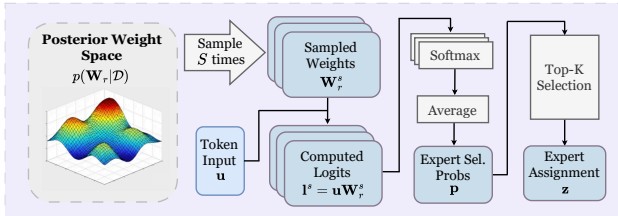

*Figure 4.* **Weight-Space Inference.** Uncertainty is modeled by sampling global parameters $\mathbf{W}_r \sim p(\mathbf{W}_r \mid \mathcal{D})$.

and latents: [2]

$$
p(\mathbf{y}^*|\mathbf{u}^*, \mathcal{D})
$$
$$
= \underbrace{\int\int\int\int p(\mathbf{y}^*|\mathbf{u}, \mathbf{z}, \mathbf{p}, \Theta)\, p(\mathbf{z}|\mathbf{p})\, p(\mathbf{p}|\mathbf{l})\, p(\mathbf{l}|\mathbf{u}^*, \Theta)\, d\mathbf{l}\, d\mathbf{p}\, d\mathbf{z}}_{\text{Latent Activation Marginalisation}}
$$
$$
\times \underbrace{p(\Theta \mid \mathcal{D})\, d\Theta}_{\text{Weight Posterior Integration}}
$$

(4)

Standard MoE training performs Point Estimation (MAP/MLE), effectively approximating the weight posterior with delta functions at $\hat{\Theta}$. However, even with fixed weights, the inner integral over the latent dependency ($\mathbf{l} \rightarrow \mathbf{p} \rightarrow \mathbf{z}$) remains. Standard routing ignores this by taking a single deterministic path (Top-K), thereby discarding the predictive uncertainty.

### 2.2. The Weight-Space Critique

The conventional approach to intractable posterior inference is to approximate the distribution over global parameters, $p(\Theta \mid \mathcal{D})$, often referred to as *Weight-Space* methods (e.g., MC Dropout (Gal & Ghahramani, 2016), SWAG (Maddox et al., 2019), Deep Ensembles (Lakshminarayanan et al., 2017)). In the context of MoE routing, this corresponds to approximating the *outer* integral in Equation 4 while ignoring the inner marginalisation over latent states. While theoretically sound, weight-space inference is indirect for the task of routing. Ideally, one might target the discrete selection $\mathbf{z}$ directly, however, this is combinatorially intractable

---

[2]We're using $\int$ to denote marginalisation over both continuous and discrete variables ($\mathbf{z}$) for notational clarity.

and insufficient, as the final output $\mathbf{y}$ also relies on the continuous probabilities $\mathbf{p}$ for expert weighting. Consequently, our primary interest lies in modelling a posterior over the *routing decision variables*—specifically the logits $\mathbf{l}$ and probabilities $\mathbf{p}$—which serve as the differentiable precursors to both selection and weighting. Weight-space methods, by contrast, rely on linear projections to propagate parameter noise from $\mathbf{W}_r$ onto these variables. This induces an indirect posterior over the routing decision variables which makes prior selection challenging and may result in an undesirable uncertainty profile. Instead of modeling the indirect cause (weight uncertainty), it is more effective to directly model the proximal effect (routing decision uncertainty).

## 3. Logit-Space Variational Inference

To overcome the structural bottlenecks of weight-space inference, we propose treating the routing logits $\mathbf{l} \in \mathbb{R}^N$ not as deterministic values, but as stochastic latent variables governed by the input $\mathbf{u}$. Our goal is to approximate the intractable posterior $p(\mathbf{l}|\mathbf{u})$ using a variational distribution $q_\phi(\mathbf{l}|\mathbf{u})$ parameterised by a lightweight inference network $\phi$. This constitutes an **amortised inference** approach, where the global network $\phi$ learns to map any input token directly to the *sufficient statistics* of its approximate posterior. We adopt the standard variational approach with multivariate Gaussian posteriors i.e., optimizing the Evidence Lower Bound (ELBO):

$$\mathcal{L}_{\text{ELBO}} = \mathbb{E}_{q_\phi(\mathbf{l}|\mathbf{u})}[\log p(\mathbf{y}|\mathbf{l}, \mathbf{u})] - \beta D_{\text{KL}}(q_\phi(\mathbf{l}|\mathbf{u}) \,||\, p(\mathbf{l}|\mathbf{u}))$$

(5)

Note that the *reconstruction term* $p(\mathbf{y}|\mathbf{l}, \mathbf{u})$ remains conditioned on the input $\mathbf{u}$, as the final MoE output depends on both the expert selection (derived from $\mathbf{l}$) and the experts' computation on the token itself. The second term is the *KL divergence term*, which regularises the posterior against a prior belief $p(\mathbf{l}|\mathbf{u})$ whose strength is governed by $\beta$.

### 3.1. Residual Learning & The Centred Prior

A critical challenge in fine-tuning is preserving the routing performance of the pre-trained model. To ensure stability, we employ a *residual learning* strategy. Instead of predicting the posterior mean from scratch, the inference network learns a *residual correction* $\Delta\boldsymbol{\mu}_\phi(\mathbf{u})$ added to the deterministic logits:

$$\boldsymbol{\mu}_{post}(\mathbf{u}) = \mathbf{l}_{det} + \Delta\boldsymbol{\mu}_\phi(\mathbf{u}), \quad \text{where } \mathbf{l}_{det} = \mathbf{u}\mathbf{W}_r \quad (6)$$

This architectural choice induces what we term as a *centred prior*: a Gaussian prior, $p$, centered on the deterministic solution: $p(\mathbf{l}|\mathbf{u}) = \mathcal{N}(\mathbf{l}_{det}, \mathbf{I})$. This formulation offers a significant theoretical advantage: by centering the prior on the deterministic baseline, the KL divergence simplifies to the distance between the learned residual and zero, that is:

$$D_{\text{KL}}(q_\phi \,||\, p) = D_{\text{KL}}(\mathcal{N}(\Delta\boldsymbol{\mu}_\phi(\mathbf{u}), \boldsymbol{\Sigma}_{post}) \,||\, \mathcal{N}(\mathbf{0}, \mathbf{I})) \quad (7)$$

### 3.2. Architecture: Variational Gaussian Logit Router

We term this framework the **Variational Gaussian Logit Router (VGLR)**, illustrated in Figure 5. The architecture employs a heteroscedastic network: a common backbone extracts features from the input token, which then branch into distinct heads to predict the residual mean $\Delta\boldsymbol{\mu}_\phi(\mathbf{u})$ and the covariance parameters.

**The Correlation Challenge** Standard variational approaches typically adopt a *mean-field (MF)* approximation, assuming the latent dimensions are statistically independent. While computationally efficient ($O(N)$), this assumption is suboptimal for routing because expert suitabilities are often highly correlated (e.g., experts specialising in similar domains). Unlike high-dimensional image latent spaces, the MoE routing space are relatively small ($N \leq 64$). This allows us to move beyond the naive mean-field baseline and explicitly model expert correlations via *full covariance (FC)* modelling. We propose a *full-covariance* formulation where the inference network predicts the entries of a lower-triangular Cholesky factor $\mathbf{L}_\phi(\mathbf{u})$, defining a dense covariance matrix $\boldsymbol{\Sigma}_{post} = \mathbf{L}\mathbf{L}^\top$. Sampling is performed via the multivariate reparameterisation trick:

$$\mathbf{l} = \mathbf{l}_{det} + \Delta\boldsymbol{\mu}_\phi(\mathbf{u}) + \mathbf{L}_\phi(\mathbf{u})\boldsymbol{\epsilon}, \quad \boldsymbol{\epsilon} \sim \mathcal{N}(\mathbf{0}, \mathbf{I}) \quad (8)$$

By capturing the off-diagonal terms in $\boldsymbol{\Sigma}_{post}$, VGLR-FC models the joint distribution of expert suitability, providing a richer uncertainty signal than deterministic or mean-field baselines. Although this incurs $O(N^2)$ complexity, it remains negligible for standard expert counts. The analytic KL loss is defined as:

$$D_{\text{KL}}(q_\phi \,||\, p) = \frac{1}{2} \left( \text{tr}(\boldsymbol{\Sigma}_{post}) + \|\Delta\boldsymbol{\mu}\|_2^2 - N - \log|\boldsymbol{\Sigma}_{post}| \right)$$

(9)

where the log-determinant term $\log|\boldsymbol{\Sigma}_{post}|$ is efficiently computed as $2\sum \log(\text{diag}(\mathbf{L}))$.

### 3.3. Inference Strategy

While the training procedure resembles a standard VAE, the inference strategy requires adaptation because the posterior predictive distribution over the final decision space involves a non-linear Softmax transformation. Since the integral of a Gaussian passing through a Softmax is intractable, we rely on Monte Carlo (MC) sampling.

**Training** We use a single sample to compute reconstruction loss, leveraging mini-batch stochasticity for efficiency.

**Inference** To obtain a robust estimate of the routing distribution, we draw $S$ samples $\{\mathbf{l}^{(s)}\}_{s=1}^S$ from the approximate posterior $q_\phi(\mathbf{l}|\mathbf{u})$. The averaged routing probabilities are computed via Monte Carlo approximation: $\mathbf{p} = \frac{1}{S}\sum_{s=1}^S \text{Softmax}(\mathbf{l}^{(s)})$. These averaged probabilities are then passed to the Top-$K$ operator to yield the final

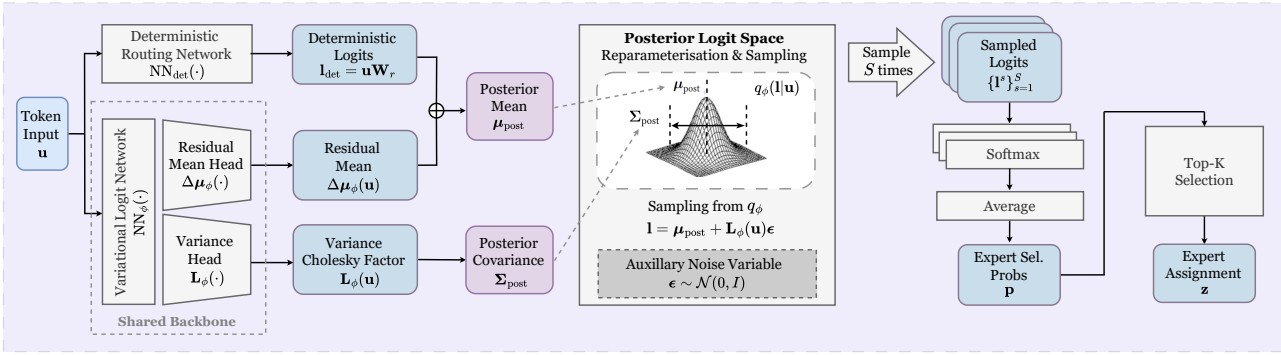

*Figure 5.* **Variational Gaussian Logit Router (VGLR) Architecture.** Latent logits are sampled from a posterior parameterised by a shared-trunk network using the reparameterisation trick, then averaged to marginalise routing uncertainty.

discrete expert assignment $\mathbf{z}$. [3] This process effectively marginalises over the uncertainty in the logit space, ensuring that decisions are robust to noise in the router's internal state.

## 4. Selection-Space Variational Inference

While Logit-Space inference (VGLR) provides a rich uncertainty representation, it relies on Monte Carlo averaging ($S > 1$) to approximate the high-dimensional posterior integral, creating a computational bottleneck for latency-critical applications. It may be more efficient to directly modeling the discrete posterior $p(\mathbf{z}|\mathbf{u})$, but this is combinatorially intractable due to the explosive search space $\binom{N}{k}$. Thus, we seek an efficient mechanism that captures the *magnitude* of uncertainty directly on the decision boundary without the overhead of multiple samples or full covariance modeling.

To address this, we propose an Bayesian approach to *temperature scaling*. Rather than learning a generic posterior, we restrict variational family $q_\phi$ to a physically meaningful 1D sub-manifold: the trajectory of distributions defined by scaling the fixed deterministic logits with a latent temperature, as in Figure 6. By learning an input-dependent temperature $T_\phi(\mathbf{u})$, we can effectively reshape the probability simplex, allowing the use of a stochastic **Sample-K** operator to introduce necessary variance into selection space.

### 4.1. Variational Formulation

We define the variational posterior $q_\phi(\mathbf{p}|\mathbf{u})$ not as a generic distribution over the simplex, but as a specific point estimate constrained to a 1D manifold. This manifold is parameterised by the fixed deterministic logits $\mathbf{l}_{\text{det}}$ and governed by a latent, input-dependent temperature $T_\phi(\mathbf{u}) \in \mathbb{R}^+$:

---

[3]Applying the non-linear Top-$K$ operator after averaging introduces a deliberate estimation bias to preserve strict $K$-expert sparsity. We provide a complete theoretical discussion of this design choice in Appendix A.3.

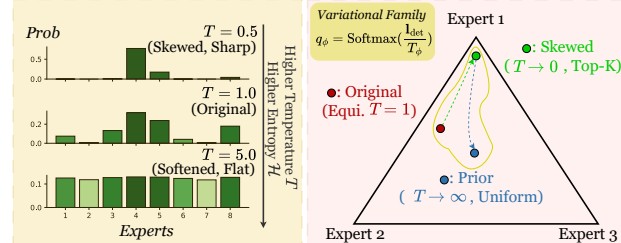

*Figure 6.* **Geometry of Variational Temperature Scaling.** The temperature $T$ acts as a radial coordinate on the simplex. A low $T$ (green) pushes the distribution toward the vertices (certainty), while a high $T$ (blue) pulls it toward the centre (uniformity). By learning $T_\phi(\mathbf{u})$, the model dynamically expands or contracts its *variational search space* (yellow region) based on input ambiguity.

$q_\phi(\mathbf{p}|\mathbf{u}) = \text{Softmax}\left(\mathbf{l}_{\text{det}}/T_\phi(\mathbf{u})\right)$. As shown in Figure 6, by varying $T_\phi$, the model can slide the posterior $\mathbf{p}$ along the trajectory connecting the sharp deterministic solution ($T \to 0$) to the maximum-entropy centre of the simplex ($T \to \infty$). The final discrete decision is obtained by applying the *Sample-K* operator to this posterior probability vector. We note that as $T_\phi(\mathbf{u}) \to 0$, the distribution $\mathbf{p}_\phi$ approaches a one-hot vector, and the Sample-K operation mathematically converges to the deterministic Top-K selection.

**Epistemic Prior** To formalise the regularisation, we define the prior $p(\mathbf{p})$ as the *uniform categorical distribution* over the $N$ experts: $p(\mathbf{p}) = \left[\frac{1}{N}, \frac{1}{N}, \ldots, \frac{1}{N}\right]$. This prior represents a state of "maximum epistemic uncertainty." It reflects the Bayesian belief that in the absence of strong evidence (e.g., OoD data or ambiguous tokens), the router should revert to a balanced, unbiased allocation rather than arbitrarily favouring a specific expert.

**Entropy as a Variational Objective** As previously discussed, entropy regularization emerges in our framework as a consequence of posterior inference. Our objective, $D_{\text{KL}}(q_\phi \,\|\, p) = \sum_{k=1}^{N} q_{\phi,k} \log q_{\phi,k} + \log N$ can be writ-

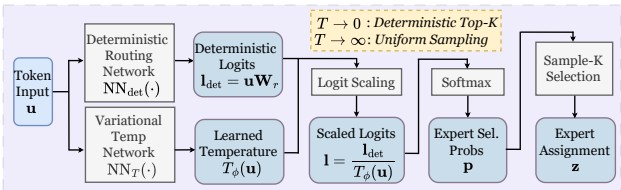

*Figure 7.* **Variational Temperature Scaling Router (VTSR).** Learn a heteroscedastic temperature via neural network and sampling from a scaled probability for expert selection.

ten in terms of the Shannon entropy: $D_{\mathrm{KL}}(q_\phi \,\|\, p) = -\mathcal{H}(q_\phi) + \text{const}$ Thus, minimising the KL divergence is mathematically equivalent to *maximising the entropy* of the routing policy. This provides a Bayesian justification for entropy penalties in sparse routing as implicit priors.

### 4.2. Variational Temperature Scaling Router (VTSR)

The architecture of the VTSR is depicted in Figure 7. It consists of a lightweight MLP that runs in parallel with the deterministic router to predict the scalar temperature $T_\phi(\mathbf{u})$. As computing gradients for Shannon entropy can be unstable, we employ a proxy objective, namely a regularisation term $\mathcal{L}_{\mathrm{reg}} = -\log T_\phi(\mathbf{u})$. This formulation shares conceptual similarities with variational tempering (Mandt et al., 2016), which introduces a latent temperature variable to dynamically anneal the variational objective across individual data points. In our context, this pushes the router toward higher stochasticity (the prior) unless the data loglikelihood (reconstruction term) strongly demands a sharp, deterministic decision. One minor downside is that the discrete sampling step requires us to use *Gumbel-Softmax* (Jang et al., 2016) to compute gradients for the learning process.

## 5. Experiments

We evaluate our proposed varational routing methods across three dimensions: (1) In-Distribution (ID) reliability, (2) Out-of-Distribution (OoD) detection capability, and (3) robustness to input perturbations. Finally, we analyse the efficiency of our approach to demonstrate practicality.

### 5.1. Experimental Setup

We evaluate VMoER across three dimensions: In-Distribution (ID) reliability, Out-of-Distribution (OoD) detection, and routing stability. This section details the models, datasets, and metrics used to ensure a rigorous comparison.

**Models and Baselines** To verify the generalisability of our approach, we conduct experiments on three distinct Mixture-of-Experts architectures varying in scale: *Granite (3B)* (IBM, 2024), *Qwen (2.7B)* (Qwen, 2024) and *DeepSeek (16B)* (Dai et al., 2024). Our proposed methods

are applied as lightweight fine-tuning strategies on top of these pre-trained backbones. We compare our two proposed families of VMoER, Logit-Space (*VGLR-MF*, *VGLR-FC*) and Selection-Space (*VTSR*), against three distinct classes of baselines. First, we establish a deterministic baseline using the standard *MAP* Top-K router. Second, we evaluate a heuristic baseline (*Temp-Scale*) which uses a fixed global temperature hyperparameter. Finally, we compare against Weight-Space Bayesian baselines, specifically MCDropout (*MCDR*) and SWAG (*SWAGR*).

**Datasets and Tasks** All evaluations are performed on Multiple-Choice Question Answering (MCQA) tasks under two distinct settings. For *in-distribution* analysis, we evaluate calibration and performance on four datasets treated as independent tasks: *OpenBookQA (OBQA)* (Mihaylov et al., 2018), *ARC-Challenge* (Clark et al., 2018), *SciQ* (Welbl et al., 2017), and *MedMCQA* (Pal et al., 2022). For *out-of-distribution* (OoD) evaluation, models are fine-tuned solely on *OBQA* (source) and evaluated on target domains of increasing semantic distance. We define near-domain shifts using *ARC-Easy* (Clark et al., 2018) and *ARC-Challenge*, and far-domain shifts using *MedMCQA* and *MMLU-Law*.

**Evaluation Metrics** We employ a comprehensive suite of metrics to assess reliability. For performance and calibration, we report *Accuracy*, *Negative Log-Likelihood (NLL)* and *Expected/Maximum Calibration Error (ECE, MCE)*. For OoD detection, we treat the task as binary classification (ID vs. OoD) and report *AUROC* and *AUPRC*. Crucially, we compare standard gating entropy signal against our proposed internal routing variance signal. Finally, routing stability is quantified by *Jaccard Similarity* of expert selection under input perturbation.

### 5.2. ID Performance and Calibration

As illustrated in Table 1, the deterministic MAP baseline suffers from significant miscalibration, characterised by high ECE scores (e.g., 0.252 on Granite). This confirms that standard top-$k$ routers tend to be overconfident.

Proposed VMoER effectively mitigate this issue. *VGLR-FC* achieves the lowest calibration error across the majority of settings, reducing ECE by over an order of magnitude on Granite ($0.252 \rightarrow 0.015$) and achieving competitive NLL scores. This suggests that modeling the full covariance of the routing distribution allows the network to capture complex expert correlations that mean-field approaches or simple dropout masks miss.

Notably, while the heuristic *Temp-Scale* baseline improves calibration, its impact on predictive performance is inconsistent across architectures. While it improves accuracy on Qwen ($0.804 \rightarrow 0.817$), it causes a significant degradation on Granite ($0.746 \rightarrow 0.716$). This accuracy degradation oc-

*Table 1.* **Main Results (OpenBookQA).** Comparison of calibration and predictive performance across three MoE architectures on the OBQA commonsense reasoning task. Results are averaged over 5 runs. Our VGLR-FC method consistently minimises calibration error (ECE) while maintaining high accuracy. VTSR also achieves strong reliability improvements. Full results in Appendix D.2.

| Method | Type | Granite-MoE | | | | Qwen-MoE | | | | DeepSeek-MoE | | | |
|--------|------|------|------|------|------|------|------|------|------|------|------|------|------|
| | | ACC ↑ | NLL ↓ | ECE ↓ | MCE ↓ | ACC ↑ | NLL ↓ | ECE ↓ | MCE ↓ | ACC ↑ | NLL ↓ | ECE ↓ | MCE ↓ |
| MAP | Det. | **0.746** | 1.38 | 0.252 | 0.472 | 0.804 | 1.37 | 0.127 | 0.273 | 0.802 | 1.120 | 0.168 | 0.321 |
| Temp-Scale | Heur. | 0.716 | 0.773 | 0.107 | 0.201 | 0.817 | 1.12 | 0.102 | 0.224 | **0.808** | 0.870 | 0.146 | 0.402 |
| MCDropout | Wei. | 0.734 | **0.650** | 0.037 | 0.298 | 0.798 | 0.982 | 0.064 | 0.152 | 0.794 | 0.729 | 0.103 | 0.302 |
| SWAG | Wei. | 0.736 | 0.652 | 0.041 | 0.290 | 0.782 | 0.912 | 0.098 | 0.146 | 0.786 | 0.665 | 0.140 | 0.288 |
| VGLR-MF | Log. | 0.742 | 0.654 | 0.026 | 0.293 | 0.796 | 0.724 | 0.028 | 0.124 | 0.788 | 0.476 | 0.067 | 0.174 |
| **VGLR-FC** | Log. | 0.740 | 0.652 | **0.015** | **0.152** | 0.802 | 0.694 | **0.014** | **0.098** | 0.800 | 0.447 | **0.054** | **0.146** |
| VTSR | Sel. | 0.736 | 0.667 | 0.052 | 0.293 | **0.818** | **0.682** | 0.022 | 0.106 | 0.798 | **0.431** | 0.060 | 0.262 |

curs because the Temp-Scale baseline replaces the standard deterministic argmax with a stochastic sampling procedure; drawing from the temperature-scaled probabilities means the original Top-$K$ expert selection is not inherently preserved. Furthermore, this instability across different models likely stems from the reliance on a single, globally-tuned temperature scalar that cannot adapt to the varying difficulty of individual tokens. In contrast, *VTSR* by learning input-dependent temperature scaling factor consistently improves calibration by an impressive amount without sacrificing accuracy.

### 5.3. Out-of-Distribution Detection

We hypothesize that the router's internal latent dynamics capture domain shifts more effectively than standard gating entropy. To test this, we evaluate the ability of our methods to distinguish in-distribution (OBQA) from near-domain shifts (ARC) and far-domain shifts (MedMCQA, MMLU-Law) using binary classification.

**Experimental Protocol** For each input sequence, we extract uncertainty signals from the final predictive token, averaged across all Bayesian-modified layers. These scores serve as the decision threshold for detecting OoD samples. We report the Area Under the Receiver Operating Characteristic (AUROC) curve for the **Granite-MoE** architecture with full results including AUPRC in Appendix D.3.

**Uncertainty Signals** We contrast the **Gate-Ent**—defined as the entropy of the expert gating distribution—against our proposed method-specific internal signals. For MCDR, we utilize *MC-Logit-Var*, representing the variance of logit samples across stochastic passes. For VGLR, we employ *Inf-Logit-Var*, which is the trace of the inferred covariance matrix. Finally, for VTSR, we examine *Inf-Temp*, the raw predicted temperature. Detailed formulations are provided in the appendix. Further, Table 2 presents the results for the Granite architecture. We observe two distinct mechanisms by which VMoER improves detection.

**1. Total Variance as a Proxy for Correlation (VGLR).**

*Table 2.* **OoD Detection Performance (Granite-MoE).** Comparison of AUROC scores using standard gating entropy (**Gate-Ent**) versus internal router signals. **Bold** indicates the best signal.

| Method | Signal | Near-Domain | | Far-Domain | | |
|--------|--------|------|------|------|------|------|
| | | ARC-E | ARC-C | Med | Law | Avg |
| MAP | Gate-Ent | 0.612 | 0.633 | 0.679 | 0.683 | 0.652 |
| MCDR | Gate-Ent | 0.612 | 0.632 | 0.684 | 0.691 | 0.655 |
| | *MC-Logit-Var* | 0.610 | 0.677 | 0.786 | 0.793 | 0.717 |
| **VGLR-FC** | Gate-Ent | 0.615 | 0.652 | 0.692 | 0.677 | 0.659 |
| | *Inf-Logit-Var* | 0.609 | **0.709** | **0.844** | **0.834** | **0.749** |
| **VTSR** | *Gate-Ent* | **0.621** | 0.693 | 0.834 | 0.824 | 0.743 |
| | *Inf-Temp* | 0.502 | 0.498 | 0.512 | 0.523 | 0.509 |

For VGLR-FC, the internal *Inf-Logit-Var* signal consistently outperforms standard entropy, improving the average AUROC by 9 points ($0.659 \rightarrow 0.749$). Standard formulation approaches routing as a multi-label problem (selecting top-$k$) using a single softmax vector, which fails to capture inter-expert dependencies. By explicitly modeling the joint distribution, the trace of the covariance matrix captures the *total volume* of routing uncertainty—including correlations between similar experts—providing a much richer signal than the flatness of the projected probability vector.

**2. Interaction of Temperature and Entropy (VTSR).** Conversely, for VTSR, the raw *Inf-Temp* signal performs poorly (0.509), while its corresponding *Gate-Ent* achieves state-of-the-art results (0.743). This suggests that temperature alone is insufficient; it is the *interaction* between the learned global ambiguity ($T_\phi$) and the structural expert conflict (logits) that matters. By scaling the logits, the learned temperature effectively "modulates" the entropy, amplifying the uncertainty signal for ambiguous inputs in a way that the static MAP baseline cannot.

### 5.4. Stability Analysis

We revisit the motivating hypothesis that deterministic routers are inherently brittle. Though our training objective does not optimise for stability, we hypothesise that our

stochasticity aids stability. We verify this by adding Gaussian noise to input embeddings and measuring the **Jaccard Similarity** of the selected experts (see Appendix B.1 for full details and full results in Appendix D.4).

*Table 3.* **Routing Stability (Qwen-MoE).** Average Jaccard Similarity of expert selection under input perturbation.

| Noise ($\sigma$) | MAP | MCDR | VGLR-FC | VTSR |
|---|---|---|---|---|
| 0.001 | 0.938 | 0.944 | **0.956** | 0.948 |
| 0.002 | 0.924 | 0.932 | **0.946** | 0.936 |
| 0.005 | 0.614 | 0.662 | **0.672** | 0.668 |
| 0.010 | 0.532 | 0.582 | 0.612 | **0.614** |

In Table 3, we observe a clear trend: as noise intensity increases ($\sigma = 0.010$), the deterministic MAP baseline suffers a sharp drop in consistency (0.532) while both VGLR-FC and VTSR exhibit superior robustness ($> 0.61$), confirming that treating routing as a distribution rather than a point estimate stabilises routing networks.

### 5.5. Practicality: Efficiency Analysis

For Bayesian routing to be viable in large-scale foundation models, the computational overhead must be negligible relative to the base model. We analyse the theoretical complexity and empirical cost (Activation Memory and FLOPs) of our methods against weight-space baselines. Detailed derivations for all calculations are provided in Appendix E.

**Activation Memory** As detailed in Table 4, weight-space methods (MCDR, SWAGR) impose significant memory burdens when parallelised. To execute $S$ forward passes simultaneously, these methods must pre-load distinct copies of the router weights (or post-dropout masks), scaling memory usage linearly by $O(S)$. Consequently, the activation memory overhead for these baselines reaches ~2.6%. In contrast, our Variational Routing methods (VGLR, VTSR) rely on fixed, lightweight inference heads. On the Granite-3B-MoE model, VGLR-FC adds only ~1.2% to the total activation memory, a negligible footprint for modern hardware.

**Computational Cost** Weight-space baselines also scale linearly with samples $S$, requiring repeated forward passes through the router's heavy projection layer ($O(LSDN)$). Our logit-space and selection-space methods amortise this cost: the inference network runs a single pass per token ($O(DH)$). Crucially, VTSR incurs **zero** additional sampling overhead in terms of FLOPs, as its complexity is independent of $S$. This decoupling allows our methods to achieve robust uncertainty quantification with minimal latency impact. Further ablation studies justifying our layer selection strategy are provided in Appendix F.

*Table 4.* **Complexity and Overhead Analysis.** Comparison on Granite-3B-MoE ($L=10$, $S=35$, $H=D/4$).

| Method | Activation Memory | | | Computational Cost (FLOPs) | | |
|---|---|---|---|---|---|---|
| | Complexity | Params | Add. % | Complexity | GFLOPs | Add. % |
| *Weight-Space Baselines (MCDR[parallelised], SWAGR)* | | | | | | |
| All | $O(LSDN)$ | 20.9M | 2.61% | $O(LSDN)$ | 0.0208 | 2.32% |
| *VMoER (Ours)* | | | | | | |
| VGLR-MF | $O(L(DH + HN))$ | 6.2M | 0.78% | $O(L(DH + SN))$ | 0.0069 | 0.77% |
| **VGLR-FC** | $O(L(DH + HN^2))$ | 9.2M | 1.15% | $O(L(DH + HN^2 + SN^2))$ | 0.0096 | 1.07% |
| **VTSR** | $O(L(DH + H))$ | 5.9M | 0.74% | $O(L(DH + N))$ | 0.0060 | 0.67% |

## 6. Related Work

**MoE Routing and Reliability.** Top language models such as DeepSeek-AI et al. (2025) and Yang et al. (2025) demonstrate that sparse expert routing enables scalable foundation models. Shazeer et al. (2017) and Guo et al. (2017) identify early routing failures, including expert collapse and severe load imbalance. Lepikhin et al. (2020) and Fedus et al. (2022) show that deterministic Top-K routing is brittle to numerical precision and input perturbations, leading to unstable expert assignments under low-precision or noisy settings. Zoph et al. (2022) further demonstrate selection drift during fine-tuning and transfer, resulting in expert under-utilisation and degraded performance. In practice, these issues are mitigated through precision management (Lepikhin et al., 2020), explicit load-balancing losses (Shazeer et al., 2017; Fedus et al., 2022), and controlled stochasticity such as noisy or softened Top-K routing (Fedus et al., 2022; Zoph et al., 2022); however, these mechanisms remain heuristic and deterministic at inference time, and do not explicitly model uncertainty over expert selection.

**Uncertainty Quantification in LLMs.** Current uncertainty quantification methods for LLMs are largely post-hoc and output-centric. Black-box approaches elicit verbalised confidence but are often poorly calibrated or require supervised tuning (Lin et al., 2022; Kapoor et al., 2024). Logit-based metrics, such as semantic entropy (Kuhn et al., 2023; Farquhar et al., 2024), estimate uncertainty by aggregating over the final vocabulary distribution. In contrast, Variational Routing extracts a latent epistemic uncertainty signal (Kendall & Gal, 2017) directly from the model's internal decision-making process prior to token generation.

**Variational Inference in Transformers.** Prior applications of variational inference (VI) in Transformers primarily target self-attention, modeling attention scores as Gaussian processes (Chen & Li, 2023; Chen et al., 2024; Bui et al., 2024), or inject noise into weights for approximate inference (Fan et al., 2020). These methods address uncertainty in feature extraction rather than expert selection. Concurrent to our work, Dialameh et al. (2025) investigate Bayesian uncertainty for MoE architectures by applying variational inference directly over the expert network weights. While this weight-space approach provides rich uncertainty estimates within the expert pathways, it

incurs substantial computational and memory overhead. In contrast, VMoER is complementary and computationally lightweight; by applying amortised variational inference (Kingma & Welling, 2013) directly to the router's decision space, we address the brittleness and miscalibration of deterministic expert selection without the burden of weight-space posteriors. Also related is Mixture-of-Variational-Experts (Hihn & Braun, 2022), which applies VI to mitigate catastrophic forgetting in continual learning.

## 7. Conclusion

In this work, we propose **Variational Routing (VMoER)**, a framework that reformulates the Mixture-of-Experts Transformer's routing mechanism as a *probabilistic latent variable model*. We introduced targeted *variational inference* methods operating directly on the decision variables: **Logit-Space** inference (VGLR) to capture semantic expert correlations, and **Selection-Space** inference (VTSR) to learn input-dependent routing stochasticity.

Our empirical analysis yields three key findings. First, the VMoER's internal variance serves as a distinct and superior signal for Out-of-Distribution detection compared to standard predictive entropy. Second, VMoER significantly improves calibration on in-distribution tasks, mitigating the overconfidence typical of deterministic Top-K selection. Third, we demonstrate that these benefits are achievable with minimal computational overhead ($< 1\%$ FLOPs), confirming that rigorous uncertainty quantification can be achieved without compromising efficiency.

In future works we hope to address remaining limitations, particularly regarding VTSR method, which, while efficient, exhibits training instability (temperature collapse) without careful initialisation. Additionally, our current evaluation is restricted to Multiple-Choice Question Answering (MCQA) benchmarks that rely on next-token prediction uncertainty. Future works will focus on stabilising optimisation dynamics, scaling VMoER to larger 70B+ parameter architectures, and extending the framework from token-level uncertainty to holistic sequence generation tasks.

## Impact Statement

This work contributes to the development of more reliable and uncertainty-aware foundation models by introducing a scalable Bayesian framework for Mixture-of-Experts (MoE) routing. By explicitly modelling uncertainty in expert selection, our approach improves calibration, routing stability under perturbations, and out-of-distribution detection, all while incurring negligible computational overhead. These properties are particularly relevant for high-stakes or open-world deployments, where overconfident predictions and brittle internal decision-making can lead to silent failures.

At the same time, the methods proposed here address a specific class of reliability challenges related to uncertainty quantification and routing robustness in MoE architectures. They do not, on their own, prevent other important failure modes such as adversarial prompt manipulation, data poisoning, or the emergence of triggered backdoors within experts or routing networks. Moreover, while improved calibration and epistemic uncertainty signals can support safer downstream decision-making, they do not guarantee correct behavior in all deployment contexts.

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

# A. Mathematical Derivations

In this section, we provide the full derivations for the variational objectives used in both the Logit-Space (VGLR) and Selection-Space (VTSR) frameworks.

## A.1. KL Divergence for VGLR with Centred Prior

In Section 3, we stated that defining the prior $p(\mathbf{l}|\mathbf{u})$ as a Gaussian centred on the deterministic logits simplifies the KL divergence significantly. We derive this result here.

**Problem Setup**  Let the variational posterior $q_\phi(\mathbf{l}|\mathbf{u})$ be a multivariate Gaussian with mean $\boldsymbol{\mu}_{\text{post}}$ and covariance $\boldsymbol{\Sigma}_{\text{post}}$. We parameterise the posterior mean using a residual formulation:

$$\boldsymbol{\mu}_{\text{post}} = \mathbf{l}_{\text{det}} + \Delta\boldsymbol{\mu}, \quad \boldsymbol{\Sigma}_{\text{post}} = \mathbf{L}\mathbf{L}^\top \tag{10}$$

We define the prior $p(\mathbf{l}|\mathbf{u})$ as a fixed multivariate Gaussian centred on the deterministic solution with identity covariance:

$$\boldsymbol{\mu}_{\text{prior}} = \mathbf{l}_{\text{det}}, \quad \boldsymbol{\Sigma}_{\text{prior}} = \mathbf{I} \tag{11}$$

**General Gaussian KL Formula.**  The Kullback-Leibler divergence between two multivariate Gaussians $q = \mathcal{N}(\boldsymbol{\mu}_1, \boldsymbol{\Sigma}_1)$ and $p = \mathcal{N}(\boldsymbol{\mu}_0, \boldsymbol{\Sigma}_0)$ of dimension $N$ is given by:

$$D_{\text{KL}}(q \,||\, p) = \frac{1}{2}\left(\text{tr}(\boldsymbol{\Sigma}_0^{-1}\boldsymbol{\Sigma}_1) + (\boldsymbol{\mu}_0 - \boldsymbol{\mu}_1)^\top \boldsymbol{\Sigma}_0^{-1}(\boldsymbol{\mu}_0 - \boldsymbol{\mu}_1) - N + \log\frac{|\boldsymbol{\Sigma}_0|}{|\boldsymbol{\Sigma}_1|}\right) \tag{12}$$

**Substitution and Simplification**  Substituting our specific parameters $(\boldsymbol{\Sigma}_0 = \mathbf{I}, |\boldsymbol{\Sigma}_0| = 1, \boldsymbol{\Sigma}_0^{-1} = \mathbf{I})$:

1. **Trace Term:**

$$\text{tr}(\boldsymbol{\Sigma}_0^{-1}\boldsymbol{\Sigma}_1) = \text{tr}(\mathbf{I}\boldsymbol{\Sigma}_{\text{post}}) = \text{tr}(\boldsymbol{\Sigma}_{\text{post}}) \tag{13}$$

2. **Quadratic Term:** The term $(\boldsymbol{\mu}_0 - \boldsymbol{\mu}_1)$ becomes $(\mathbf{l}_{\text{det}} - (\mathbf{l}_{\text{det}} + \Delta\boldsymbol{\mu})) = -\Delta\boldsymbol{\mu}$. Thus:

$$(\boldsymbol{\mu}_0 - \boldsymbol{\mu}_1)^\top \boldsymbol{\Sigma}_0^{-1}(\boldsymbol{\mu}_0 - \boldsymbol{\mu}_1) = (-\Delta\boldsymbol{\mu})^\top \mathbf{I}(-\Delta\boldsymbol{\mu}) = \|\Delta\boldsymbol{\mu}\|_2^2 \tag{14}$$

3. **Log-Determinant Term:**

$$\log\frac{|\boldsymbol{\Sigma}_0|}{|\boldsymbol{\Sigma}_1|} = \log 1 - \log|\boldsymbol{\Sigma}_{\text{post}}| = -\log|\boldsymbol{\Sigma}_{\text{post}}| \tag{15}$$

**General Form**  Combining these terms yields the standard KL loss for a Gaussian posterior against a standard normal prior:

$$D_{\text{KL}}(q_\phi \,||\, p) = \frac{1}{2}\left(\text{tr}(\boldsymbol{\Sigma}_{\text{post}}) + \|\Delta\boldsymbol{\mu}\|_2^2 - N - \log|\boldsymbol{\Sigma}_{\text{post}}|\right) \tag{16}$$

**Parameterisation via Cholesky Factor**  In the Full-Covariance (VGLR-FC) variant, we parameterise the covariance matrix using its Cholesky decomposition $\boldsymbol{\Sigma}_{\text{post}} = \mathbf{L}\mathbf{L}^\top$, where $\mathbf{L}$ is a lower triangular matrix with positive diagonal entries. Substituting this into the general equation:

1. **Trace Term:** Using the cyclic property of the trace and the definition of the Frobenius norm:

$$\text{tr}(\boldsymbol{\Sigma}_{\text{post}}) = \text{tr}(\mathbf{L}\mathbf{L}^\top) = \sum_{i,j} L_{ij}^2 = \|\mathbf{L}\|_F^2 \tag{17}$$

2. **Log-Determinant Term:** The determinant of a triangular matrix is the product of its diagonal entries:

$$\log|\boldsymbol{\Sigma}_{\text{post}}| = \log|\mathbf{L}\mathbf{L}^\top| = \log(|\mathbf{L}||\mathbf{L}|) = 2\log\left(\prod_{i=1}^N L_{ii}\right) = 2\sum_{i=1}^N \log L_{ii} \tag{18}$$

Substituting these back yields the final computational objective used in our implementation (Algorithm 2):

$$D_{\text{KL}}(q_\phi \,||\, p) = \frac{1}{2} \left( \|\Delta\boldsymbol{\mu}\|_2^2 + \|\mathbf{L}\|_F^2 - 2\sum_{i=1}^{N} \log L_{ii} - N \right) \tag{19}$$

This formulation is numerically stable and confirms that minimising the KL divergence drives the residual $\Delta\boldsymbol{\mu} \to \mathbf{0}$ and the Cholesky factor $\mathbf{L} \to \mathbf{I}$ (since $\|\mathbf{I}\|_F^2 = N$ and $2\sum \log 1 = 0$, the minimum is zero).

## A.2. Proof of Entropy Equivalence for VTSR

In Section 4, we claimed that for the Selection-Space Router (VTSR), minimising the KL divergence between the posterior and a uniform prior is analytically equivalent to maximising the entropy of the routing distribution.

**Problem Setup.** Let $q_\phi(\mathbf{p}|\mathbf{u})$ be the discrete variational posterior distribution (a probability vector of size $N$). Let $p(\mathbf{p})$ be the uniform categorical prior:

$$p_k = \frac{1}{N} \quad \forall k \in \{1, \ldots, N\} \tag{20}$$

**Derivation.** The definition of the KL divergence for discrete distributions is:

$$D_{\text{KL}}(q_\phi \,||\, p) = \sum_{k=1}^{N} q_{\phi,k} \log \frac{q_{\phi,k}}{p_k} \tag{21}$$

Expanding the logarithm term:

$$D_{\text{KL}}(q_\phi \,||\, p) = \sum_{k=1}^{N} q_{\phi,k} (\log q_{\phi,k} - \log p_k) \tag{22}$$

Substituting $p_k = 1/N$:

$$D_{\text{KL}}(q_\phi \,||\, p) = \sum_{k=1}^{N} q_{\phi,k} \log q_{\phi,k} - \sum_{k=1}^{N} q_{\phi,k} \log \left( \frac{1}{N} \right) \tag{23}$$

Since $\log(1/N)$ is a constant with respect to $k$, it can be pulled out of the summation. Furthermore, the sum of probabilities $\sum q_{\phi,k} = 1$:

$$D_{\text{KL}}(q_\phi \,||\, p) = \underbrace{\sum_{k=1}^{N} q_{\phi,k} \log q_{\phi,k}}_{-\mathcal{H}(q_\phi)} - \log \left( \frac{1}{N} \right) \cdot \underbrace{\sum_{k=1}^{N} q_{\phi,k}}_{1} \tag{24}$$

Recognising the definition of Shannon entropy $\mathcal{H}(q_\phi) = -\sum q_{\phi,k} \log q_{\phi,k}$, we obtain:

$$D_{\text{KL}}(q_\phi \,||\, p) = -\mathcal{H}(q_\phi) + \log N \tag{25}$$

Since $\log N$ is a constant determined solely by the model architecture (number of experts), minimising the KL divergence is mathematically identical to maximising the Shannon entropy $\mathcal{H}(q_\phi)$. $\square$

## A.3. Forward-Pass Requirements and Estimation Bias

Prompted by an insightful reviewer question regarding the theoretical nature of our estimator, we provide a formal clarification on the forward-pass requirements and the deliberate estimation bias introduced by the discrete expert selection.

As introduced in Section 3.3, our local Bayesian Model Averaging (BMA) provides a proper, unbiased Monte Carlo estimator of the continuous routing probabilities:

$$\bar{\mathbf{p}} = \frac{1}{S} \sum_{s=1}^{S} \text{Softmax}(\mathbf{l}^{(s)}) \tag{26}$$

However, with respect to the final discrete expert selection $\mathbf{z}$, this process serves as a biased approximation. Standard MoE architectures utilize a non-linear Top-$K$ masking operation. Because the expectation of a non-linear function is not equal to the function of the expectation, applying the Top-$K$ mask to our averaged probabilities introduces bias:

$$\text{Top-}K(\mathbb{E}[\mathbf{p}]) \neq \mathbb{E}[\text{Top-}K(\mathbf{p})] \tag{27}$$

This bias is a deliberate and necessary design choice to preserve the core computational efficiency of the MoE architecture. Our VMoER framework computes the **biased** expert assignment by applying the Top-$K$ mask to the averaged continuous probabilities:

$$\bar{\mathbf{z}}_{\text{biased}} = \text{Top-}K(\bar{\mathbf{p}}) = \text{Top-}K\left(\frac{1}{S}\sum_{s=1}^{S}\text{Softmax}(\mathbf{l}^{(s)})\right) \tag{28}$$

In contrast, a strictly **unbiased** estimator would evaluate the Top-$K$ mask for every individual sample $s$ and average the resulting sparse vectors:

$$\bar{\mathbf{z}}_{\text{unbiased}} = \frac{1}{S}\sum_{s=1}^{S}\text{Top-}K(\text{Softmax}(\mathbf{l}^{(s)})) \tag{29}$$

Mathematically, this unbiased formulation could cause a single token to activate a large union of different experts across the $S$ samples. If we computed this strictly unbiased forward pass, it would completely destroy the $K$-expert sparsity bottleneck and drastically inflate the FLOP count.

By taking the expected continuous probability $\bar{\mathbf{p}}$ before applying the Top-$K$ routing, VMoER maintains strict $K$-expert sparsity while still allowing the Bayesian uncertainty over the logits to dictate the final ranking and selection of those experts.

## B. Full Motivational Experiment Details

This appendix provides the comprehensive experimental setup and results for the motivational studies mentioned in Section 1. Results displayed in Section 1 are the compact version of what's explained in detail below. These experiments reveal the fundamental **brittleness** of standard deterministic routing under perturbation and demonstrate the potential of stochasticity to improve calibration. Furthermore, as modern MoE LLMs are deep architectures, we analyse these effects layer-by-layer to identify which components are most sensitive.

### B.1. Experiment I: Brittleness of Deterministic Routing

We investigate the hypothesis that if a router has learned a robust mapping, its decisions should be stable under minimal, non-semantic perturbations. A significant shift in expert selection due to negligible noise indicates inherent brittleness.

**Methodology**  The experiment is conducted on the **Granite-MoE-3B** model, fine-tuned on the **OpenBookQA** dataset. Since the model possesses a deep, stacked architecture, we inject noise independently at each MoE layer to isolate local sensitivity and prevent the cascading of perturbations from earlier layers. Regarding the injection point, we introduce the perturbation immediately prior to the self-attention layer. This ensures that the token representations are perturbed just before they are processed as a sequence and fed into the router, allowing us to directly measure the router's sensitivity to input variations (as illustrated in Figure 8).

For each token embedding $\mathbf{x}$, a perturbed version $\mathbf{x}'$ is generated by adding Gaussian noise:

$$\mathbf{x}' = \mathbf{x} + \boldsymbol{\epsilon}, \quad \text{where } \boldsymbol{\epsilon} \sim \mathcal{N}(\mathbf{0}, \sigma^2\mathbf{I}) \tag{30}$$

To ensure the noise is non-semantic, the standard deviation $\sigma$ is set proportional to the average L2 norm of the token embeddings, $\bar{L}$. We test multiple noise levels defined by a scaling factor $\gamma$:

$$\sigma = \gamma \cdot \bar{L}, \quad \text{where } \gamma \in \{0.001, 0.002, 0.005, 0.007, 0.01, 0.02, 0.05\} \tag{31}$$

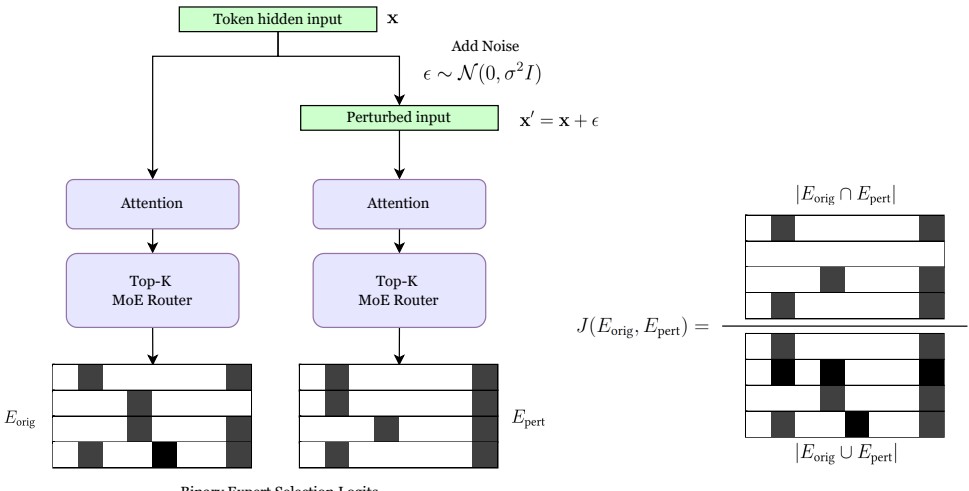

*Figure 8.* **Experimental setup** for quantifying the brittleness of deterministic routing.

For each token and noise level $\gamma$, we compare the set of $K$ experts selected for the original input ($E_{\text{orig}}$) vs. the perturbed input ($E_{\text{pert}}$) using the **Jaccard Similarity**:

$$J(E_{\text{orig}}, E_{\text{pert}}) = \frac{|E_{\text{orig}} \cap E_{\text{pert}}|}{|E_{\text{orig}} \cup E_{\text{pert}}|} \tag{32}$$

**Results: General Instability**    Figure 9 (a) presents the mean Jaccard similarity across all layers. We observe that even minute noise ($\gamma \geq 0.005$) causes a sharp drop in stability. Based on these sweeps, we select $\gamma = 0.01$ as the diagnostic noise level for detailed layer-wise analysis, as it is sensitive enough to reveal instability without saturating the effect. Figure 1a is the abbreviated version of Figure 9 (a) averaged across all MoE layers.

**Layer-wise Analysis**    Figure 9 (b) details the distribution of stability scores at each layer ($\gamma = 0.01$). The brittleness is not uniform; it is concentrated in specific regions:

- **Input Layers (0-1):** Highly unstable, likely due to the raw nature of input embeddings.

- **Transition Layers (5-8, 19-20, 28-31):** We observe significant instability at these depths, suggesting that routing decisions at network transition points are particularly fragile.

The layer-wise analysis also inform us location of the most susceptible layers, which will be the targeted layers we Bayesianfy in later experiments.

### B.2. Experiment II: Potentials of Stochastic Routing

We next investigate whether introducing simple, ad-hoc stochasticity can improve model behaviour. If random noise proves beneficial, it motivates a principled Bayesian framework that learns this stochasticity from data.

**Methodology**    We modify the expert selection mechanism of a single MoE layer at a time. Instead of the deterministic Top-K selection, we employ a **Temperature Sampling** strategy (Figure 10). Logits are scaled by a temperature $T$, and $K$ experts are sampled without replacement from the resulting softmax distribution $\mathbf{p} = \text{softmax}(\text{logits}/T)$.

**Results: Calibration vs. Accuracy**    These experiments are conducted on the pre-trained base version of **Granite-MoE-3B** without task-specific fine-tuning. Figure 11 displays the impact on Accuracy (ACC) and Expected Calibration Error (ECE) across different temperatures.

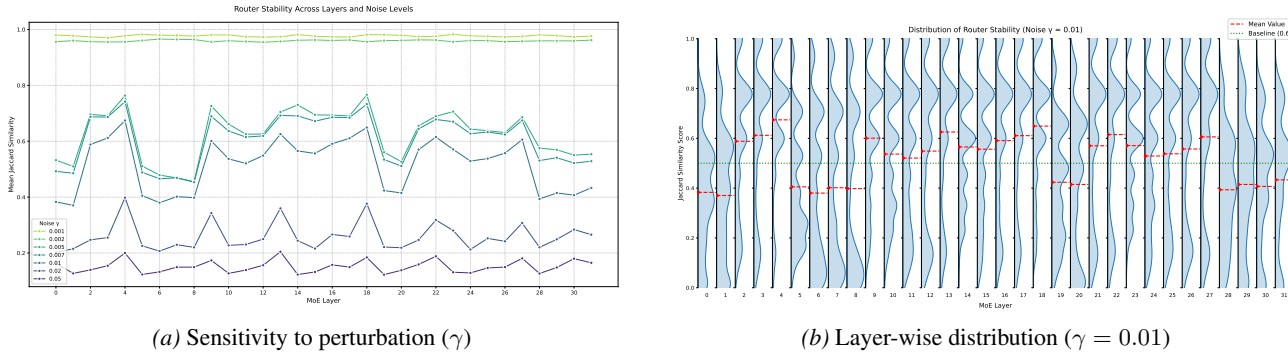

*(a)* Sensitivity to perturbation ($\gamma$)          *(b)* Layer-wise distribution ($\gamma = 0.01$)

*Figure 9.* **Brittleness Analysis Results.** (a) Mean Jaccard similarity across layers shows high sensitivity to minute input perturbations. (b) At a fixed diagnostic noise level ($\gamma = 0.01$), instability is not uniform but concentrated in specific layer groups (e.g., 0-1, 28-31).

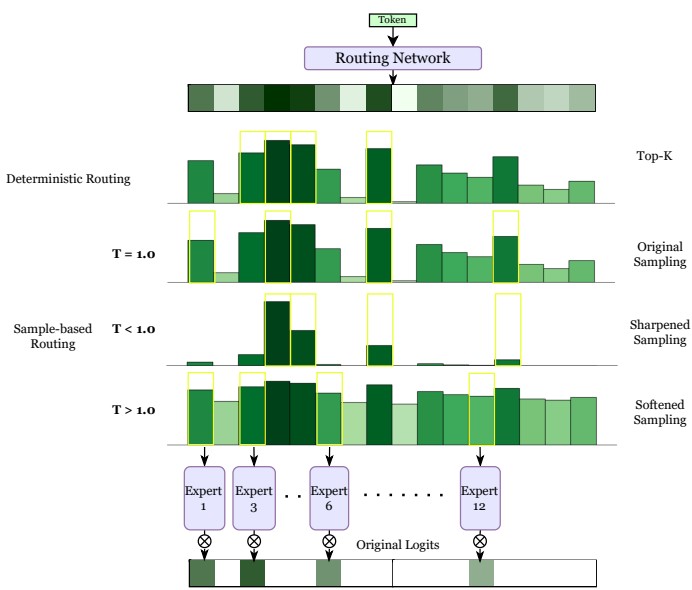

*Figure 10.* Experimental setup for introducing stochastic routing at a single MoE layer.

1. **Early Layer Sensitivity:** Stochasticity in the first two layers causes significant accuracy degradation, confirming they require stable, deterministic feature extraction.

2. **Calibration Gain:** In middle and later layers, stochasticity (particularly $T = 0.3$) consistently reduces ECE (improves calibration) while maintaining accuracy. This suggests that "softening" the router acts as an effective regulariser against overconfidence.

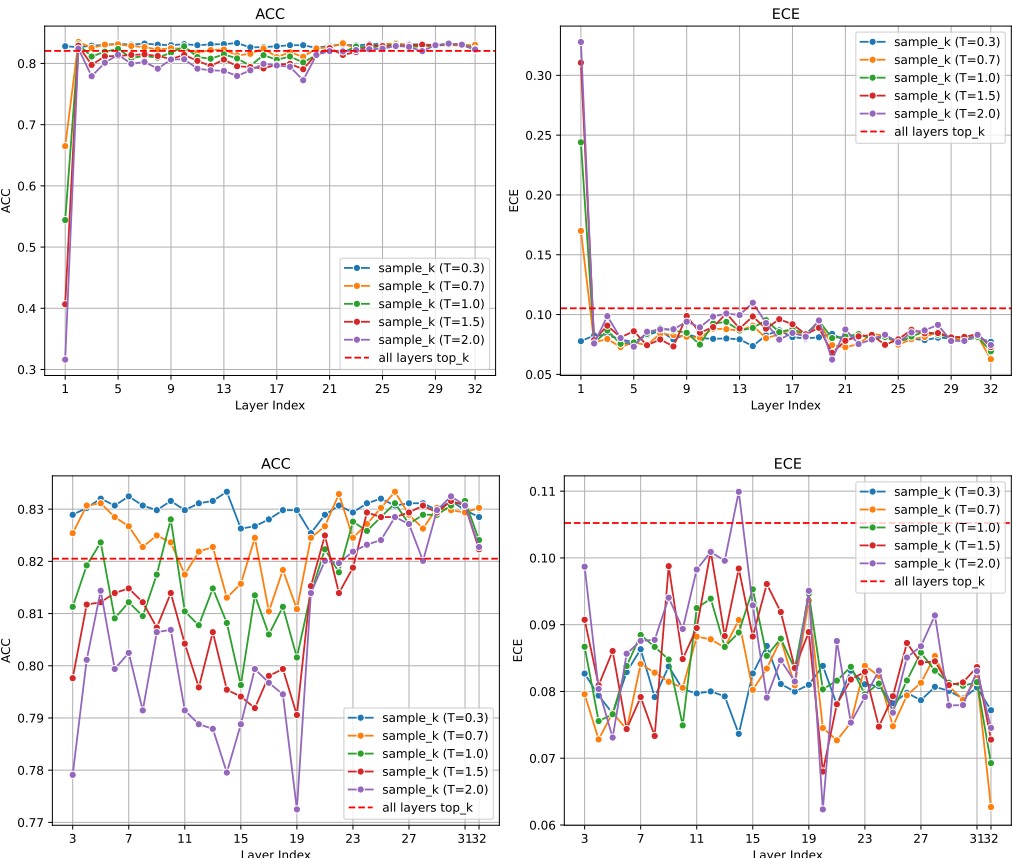

*Figure 11.* Impact of stochastic routing on Accuracy (left) and ECE (right). The top row shows all layers; the bottom row excludes layer 0 to highlight trends in deeper layers. Stochasticity often improves calibration (lower ECE) without harming accuracy in later layers.

**Conclusion** These findings provide the empirical foundation for our method: simple randomness improves calibration but hurts early layers. This motivates the **Variational MoE** approach, which learns *where* and *how much* stochasticity to apply, rather than relying on global heuristics.

## C. VMoER Implementation Details

This appendix provides the low-level implementation details required to reproduce VMoER framework. We present the formal algorithmic pseudocode, a reference PyTorch implementation, and the specific architecture hyperparameters used in our experiments.

### C.1. Algorithmic Pseudocode

Algorithms 1, 2 and 3 detail the forward pass and training logic for the Logit-Space (VGLR) and Selection-Space (VTSR) variants, respectively.

**Algorithm 1** VGLR-MF (Mean-Field)

**Require:** Input $\mathbf{u}$, Target $\mathbf{y}_{\text{true}}$, Inference Network $\phi$
1: Compute deterministic logits: $\mathbf{l}_{\text{det}} \leftarrow \mathbf{u}\mathbf{W}_r$
2: Predict posterior params: $\Delta\boldsymbol{\mu}, \boldsymbol{\sigma} \leftarrow \Delta\boldsymbol{\mu}_\phi(\mathbf{u}), \boldsymbol{\sigma}_\phi(\mathbf{u})$
3: Set posterior mean: $\boldsymbol{\mu}_{\text{post}} \leftarrow \mathbf{l}_{\text{det}} + \Delta\boldsymbol{\mu}$
   **Training Phase:**
4: Sample Noise: $\boldsymbol{\epsilon} \sim \mathcal{N}(0, I)$
5: Reparameterise Logit sample: $\mathbf{l} \leftarrow \boldsymbol{\mu}_{\text{post}} + \boldsymbol{\sigma} \odot \boldsymbol{\epsilon}$
6: Select Experts: $\mathbf{z} = \text{Top-K}(\text{softmax}(\mathbf{l}))$
7: Compute Task Loss: $\mathcal{L}_{\text{task}}(\mathbf{u}, \mathbf{y}_{\text{true}})$
8: Compute KL Divergence:
   $\mathcal{L}_{\text{KL}} = \frac{1}{2}\sum_i \left(\Delta\mu_i^2 + \sigma_i^2 - 2\log\sigma_i - 1\right)$
9: Optimisation: $\phi \leftarrow \phi - \eta\nabla_\phi(\mathcal{L}_{\text{task}} + \beta\mathcal{L}_{\text{KL}})$
   **Inference Phase:**
10: Sample $S$ posterior logits: $\mathbf{p}_{\text{sum}} \leftarrow \mathbf{0}$
11: **for** $s = 1$ to $S$ **do**
12:    $\boldsymbol{\epsilon}' \sim \mathcal{N}(0, I)$
13:    $\mathbf{l}^s \leftarrow \boldsymbol{\mu}_{\text{post}} + \boldsymbol{\sigma} \odot \boldsymbol{\epsilon}'$
14:    $\mathbf{p}_{\text{sum}} \leftarrow \mathbf{p}_{\text{sum}} + \text{softmax}(\mathbf{l}^s)$
15: **end for**
16: Select Experts: $\mathbf{z} = \text{Top-K}(\mathbf{p}_{\text{sum}}/S)$

**Algorithm 2** VGLR-FC (Full-Covariance)

**Require:** Input $\mathbf{u}$, Target $\mathbf{y}_{\text{true}}$, Inference Network $\phi$
1: Compute deterministic logits: $\mathbf{l}_{\text{det}} \leftarrow \mathbf{u}\mathbf{W}_r$
2: Predict posterior params: $\Delta\boldsymbol{\mu}, \mathbf{L} \leftarrow \Delta\boldsymbol{\mu}_\phi(\mathbf{u}), \mathbf{L}_\phi(\mathbf{u})$
3: Set posterior mean: $\boldsymbol{\mu}_{\text{post}} \leftarrow \mathbf{l}_{\text{det}} + \Delta\boldsymbol{\mu}$
   **Training Phase:**
4: Sample Noise: $\boldsymbol{\epsilon} \sim \mathcal{N}(0, I)$
5: Reparameterise Logit sample: $\mathbf{l} \leftarrow \boldsymbol{\mu}_{\text{post}} + \mathbf{L}\boldsymbol{\epsilon}$
6: Select Experts: $\mathbf{z} = \text{Top-K}(\text{softmax}(\mathbf{l}))$
7: Compute Task Loss: $\mathcal{L}_{\text{task}}(\mathbf{u}, \mathbf{y}_{\text{true}})$
8: Compute KL Divergence:
   $\mathcal{L}_{\text{KL}} = \frac{1}{2}\left(\|\Delta\boldsymbol{\mu}\|_2^2 + \|\mathbf{L}\|_F^2 - 2\sum_i \log L_{ii} - N\right)$
9: Optimisation: $\phi \leftarrow \phi - \eta\nabla_\phi(\mathcal{L}_{\text{task}} + \beta\mathcal{L}_{\text{KL}})$
   **Inference Phase:**
10: Sample $S$ posterior logits: $\mathbf{p}_{\text{sum}} \leftarrow \mathbf{0}$
11: **for** $s = 1$ to $S$ **do**
12:    $\boldsymbol{\epsilon}' \sim \mathcal{N}(0, I)$
13:    $\mathbf{l}^s \leftarrow \boldsymbol{\mu}_{\text{post}} + \mathbf{L}\boldsymbol{\epsilon}'$
14:    $\mathbf{p}_{\text{sum}} \leftarrow \mathbf{p}_{\text{sum}} + \text{softmax}(\mathbf{l}^s)$
15: **end for**
16: Select Experts: $\mathbf{z} = \text{Top-K}(\mathbf{p}_{\text{sum}}/S)$

**Algorithm 3** Variational Temperature Sampling Router (VTSR)

**Require:** Input $\mathbf{u}$, Target $\mathbf{y}_{\text{true}}$, Inference $\phi$
1: Compute deterministic logits: $\mathbf{l}_{\text{det}} \leftarrow \mathbf{u}\mathbf{W}_r$
2: Predict Temperature: $T_\phi \leftarrow \text{Softplus}(\text{NeuralNet}_\phi(\mathbf{u})) + \epsilon_{\text{min}}$
   **Training Phase (Gumbel-Softmax):**
3: Sample Gumbel Noise: $\mathbf{g} \sim \text{Gumbel}(0, 1)$
4: Relaxed Selection: $\mathbf{z}_{\text{soft}} \leftarrow \text{Top-KSoftmax}((\mathbf{l}_{\text{det}} + \mathbf{g})/T_\phi)$
5: Compute Task Loss: $\mathcal{L}_{\text{task}}(\mathbf{u}, \mathbf{y}_{\text{true}})$ *// Using soft routing weights $\mathbf{z}_{soft}$*
6: Compute Reg Loss: $\mathcal{L}_{\text{reg}} = -\log(T_\phi)$
7: Optimisation: $\phi \leftarrow \phi - \eta\nabla_\phi(\mathcal{L}_{\text{task}} + \beta\mathcal{L}_{\text{reg}})$
   **Inference Phase (Exact Sampling):**
8: Scale Logits: $\mathbf{l}_{\text{scaled}} \leftarrow \mathbf{l}_{\text{det}}/T_\phi$
9: Compute Probabilities: $\mathbf{p} \leftarrow \text{Softmax}(\mathbf{l}_{\text{scaled}})$
10: Select Experts: $\mathbf{z} \sim \text{Sample-K}(\mathbf{p})$ (Draw $K$ without replacement)

## C.2. Key Implementation Snippets

We provide the core PyTorch logic for the two primary variational routers. For clarity, we omit standard MoE boilerplate (e.g., Top-K sorting, capacity limiting) and focus on the variational inference mechanisms.

**VGLR-FC: Shared Backbone & Cholesky Decomposition** The Full-Covariance implementation features a shared backbone for parameter efficiency and a Cholesky parameterisation to ensure valid covariance matrices.

```
1  class FullCovarianceVariationalRouter(MoERouter):
2      def __init__(self, config, existing_router):
3          super().__init__(config, existing_router)
4          # 1. Frozen Base Router (The "Prior" Mean)
5          self.mean_base = nn.Linear(config.hidden_size, config.num_experts, bias=False)
6          self.mean_base.load_state_dict(existing_router.layer.state_dict())
7          self.mean_base.requires_grad_(False)
```

```python
        # 2. Shared Uncertainty Backbone (Trainable)
        # Extracts features relevant for both shift (mean) and correlation (covariance)
        self.backbone = nn.Sequential(
            nn.Linear(config.hidden_size, config.hidden_size // 4, bias=False),
            nn.ReLU()
        )

        # 3. Heads
        # Head A: Residual Mean (Shift)
        self.mean_head = nn.Linear(config.hidden_size // 4, config.num_experts, bias=False
    )

        # Head B: Cholesky Factor (Correlation)
        num_chol = config.num_experts * (config.num_experts + 1) // 2
        self.cholesky_head = nn.Linear(config.hidden_size // 4, num_chol, bias=False)

        # Init Cholesky head near zero so we start with Identity covariance
        nn.init.normal_(self.cholesky_head.weight, mean=0.0, std=1e-3)

    def forward(self, hidden_states, training=True):
        # 1. Deterministic Path (Prior)
        with torch.no_grad():
            mu_base = self.mean_base(hidden_states)

        # 2. Variational Path (Shared Backbone)
        features = self.backbone(hidden_states)
        mu_resid = self.mean_head(features)
        flat_chol = self.cholesky_head(features)

        # 3. Construct Distribution
        L = self._build_cholesky(flat_chol)
        mu_final = mu_base + mu_resid
        logit_dist = torch.distributions.MultivariateNormal(mu_final, scale_tril=L)

        # 4. Sampling
        if training:
            logits = logit_dist.rsample()
        else:
            # Multi-sample MC approximation during inference
            samples = logit_dist.rsample((self.num_mc_samples_inference,))
            probs = torch.softmax(samples, dim=-1).mean(dim=0)
            logits = torch.log(probs.clamp(min=1e-9))

        # (Standard Top-K gating logic omitted for brevity...)
        return logits, mu_resid, L

    def _build_cholesky(self, flat_params):
        """Maps flat predictions to a lower-triangular matrix with positive diagonal."""
        B, N = flat_params.shape[0], self.num_experts
        L = torch.zeros(B, N, N, device=flat_params.device)
        tril_indices = torch.tril_indices(N, N)
        L[:, tril_indices[0], tril_indices[1]] = flat_params

        # Exponentiate diagonal to ensure strict positivity
        diag_idx = torch.arange(N)
        L[:, diag_idx, diag_idx] = torch.exp(L[:, diag_idx, diag_idx])
        return L

    def kl_divergence(self, mu, L):
        """Analytic KL[N(mu, LL^T) || N(0, I)]"""
        E = self.num_experts
        trace_term = (L**2).sum(dim=(-1, -2))
        mu_sq_term = (mu**2).sum(-1)
        log_det_term = 2 * torch.log(torch.diagonal(L, dim1=-2, dim2=-1)).sum(-1)
```

```
72          return 0.5 * (trace_term + mu_sq_term - E - log_det_term).mean()
```

*Listing 1.* PyTorch implementation of VGLR (Full Covariance) with Shared Backbone.

**VTSR: Differentiable Temperature Sampling**   The Selection-Space router relies on the Gumbel-Softmax relaxation to backpropagate gradients through the discrete expert selection process.

```
1  class VariationalTemperatureRouter(MoERouter):
2      def __init__(self, config, existing_router):
3          super().__init__(config, existing_router)
4          self.layer.requires_grad_(False) # Freeze base router
5
6          # Learnable Temperature Network
7          self.temperature_net = nn.Sequential(
8              nn.Linear(config.hidden_size, config.hidden_size // 4),
9              nn.ReLU(),
10             nn.Linear(config.hidden_size // 4, 1),
11             nn.Softplus() # Constraint: T > 0
12         )
13         self.last_temperature = None
14
15     def forward(self, hidden_states):
16         # 1. Deterministic Logits
17         with torch.no_grad():
18             logits = self.layer(hidden_states).float()
19
20         # 2. Predict Temperature
21         # Add epsilon to prevent division by zero
22         temp = self.temperature_net(hidden_states) + 1e-6
23         self.last_temperature = temp
24         scaled_logits = logits / temp
25
26         if self.training:
27             # Training: Gumbel-Softmax (Differentiable)
28             # hard=True: One-hot forward, soft backward gradients
29             gumbel_probs = F.gumbel_softmax(scaled_logits, tau=1.0, hard=True)
30             top_k_indices = torch.topk(gumbel_probs, self.top_k, dim=1).indices
31         else:
32             # Inference: Exact Categorical Sampling (Sample-K)
33             probs = torch.softmax(scaled_logits, dim=-1)
34             top_k_indices = torch.multinomial(probs, self.top_k, replacement=False)
35
36         # (Standard gating logic continues...)
37         # We return standard MoE outputs to maintain API compatibility
38         return top_k_indices, scaled_logits
39
40     def temp_loss(self):
41         """
42         Side-channel access to the regularization loss.
43         Called externally during the training loop: loss += beta * router.temp_loss()
44         """
45         if self.last_temperature is None:
46             return 0.0
47         # Minimize -log(T) <=> Maximize Entropy
48         return -torch.log(self.last_temperature).mean()
```

*Listing 2.* PyTorch implementation of VTSR.

## D. Experiment Details

### D.1. Models, Datasets, and Preprocessing

**Base Models**   We evaluate our variational routing framework on three state-of-the-art open-source MoE architectures, representing distinct design philosophies in sparse routing. Table 5 summarises the architectural details derived from the

official model checkpoints.

- **Granite-MoE (3B):** A dense-like MoE from IBM with 40 experts and 8 active per token. It uses a standard Top-K gating mechanism.

- **Qwen1.5-MoE (2.7B):** A highly sparse model with 60 fine-grained experts (4 active), known for its shared expert routing mechanism (though we only intervene on the routed experts).

- **DeepSeek-MoE (16B):** A large-scale model with 64 routed experts, employing a specialized "Shared + Routed" architecture to separate common knowledge from specialized tasks.

| Specification | Granite-MoE | Qwen1.5-MoE | DeepSeek-MoE |
|---|---|---|---|
| HuggingFace ID | `granite-3.1-3b-a800m` | `Qwen1.5-MoE-A2.7B` | `deepseek-moe-16b` |
| Total Parameters | 3.3B | 14.3B | 16.4B |
| Active Params | 800M | 2.7B | 2.8B |
| MoE Layers | 32 | 24 | 27 |
| Hidden Dimension | 1,536 | 2,048 | 2,048 |
| Routed Experts ($N$) | 40 | 60 | 64 |
| Active Experts ($K$) | 8 | 4 | 6 |

*Table 5.* Configurations of the base MoE models used in this study. All models are fine-tuned from their base instruct versions.

**Datasets & Experimental Roles**  To evaluate both calibration and Out-of-Distribution (OoD) detection, we employ a diverse suite of Multiple-Choice Question Answering (MCQA) datasets. Crucially, several datasets serve dual roles across our two main experiments:

1. **Experiment 1 (ID Calibration):** We treat **OBQA**, **ARC-C**, **SciQ**, and **MedMCQA** as individual In-Distribution tasks. For this experiment, we define specific Training/Testing splits to fine-tune and evaluate the router on the *same* domain.

2. **Experiment 2 (OoD Detection):** We treat **OBQA** as the sole source of ID knowledge (training). All other datasets are used strictly as unseen Out-of-Distribution test sets to measure the router's uncertainty under shift.

Table 6 details the exact split sizes used for each role.

| Dataset | Domain | Role 1: ID Calibration *(Train / Val / Test)* | Role 2: OoD Detection *(Shift Type)* |
|---|---|---|---|
| **OBQA** | Commonsense Science | 5000 / 50 / 500 | **ID Source** (Training) |
| **ARC-C** | Primary Science (Hard) | 2000 / 50 / 500 | **Near-OoD** (OoD-S) |
| **ARC-E** | Primary Science (Easy) | – *Not Used* – | **Near-OoD** (OoD-S) |
| **SciQ** | Broad STEM | 5000 / 50 / 500 | – *Not Used* – |
| **MedMCQA** | Professional Medicine | 5000 / 50 / 500 | **Far-OoD** (OoD-L) |
| **MMLU-Law** | Professional Law | – *Not Used* – | **Far-OoD** (OoD-L) |

*Table 6.* Benchmark datasets and their usage across experiments. Note that datasets like ARC-C and MedMCQA have explicit training splits for Exp 1 (Calibration), but are treated as held-out test sets for Exp 2 (OoD Detection) where the model is trained solely on OBQA.

**Preprocessing & Prompt Engineering**  We format all Multiple-Choice Question Answering (MCQA) tasks using a strict chat template to enforce consistent output behavior. The system instruction explicitly constrains the model to output a single character (A/B/C/D) to facilitate automated parsing.

```
1  SYSTEM_INSTRUCTION = (
2      "You are a multiple-choice quiz answer generator. "
3      "Respond with ONLY the letter of the correct option, for example, 'A', 'B', 'C', or 'D
       '."
4  )
5  chat = [
6      {"role": "system", "content": SYSTEM_INSTRUCTION},
7      {"role": "user", "content": example["question"]}
8  ]
```

*Listing 3.* Prompt engineering template for MCQA tasks.

This prompt structure is applied uniformly across all models using their respective tokenizer's `apply_chat_template` method, ensuring that performance differences are attributable to the routing mechanism rather than prompting inconsistencies.

### D.2. Experiment I: In-Distribution Performance & Training Details

In this experiment, we evaluate the capacity of the VMoER framework to improve model calibration on In-Distribution tasks without degrading predictive accuracy. Unlike the OoD settings, here the router is trained and evaluated on the same domain (using the splits defined in Table 6).

**Training Setup**  To ensure the variational routers operate on a high-quality representation space while maintaining efficiency, we employ a **Two-Stage Training Protocol** combined with a selective application strategy:

1. **Stage 1: Deterministic MAP Adaptation (3 Epochs).** We first fine-tune the base model using Low-Rank Adaptation (LoRA) to establish a strong deterministic baseline (the Maximum A Posteriori estimate). LoRA adapters are applied to the attention modules (Q/K/V projections) and the Expert networks. This stage runs for 3 epochs to ensure convergence of the task-specific weights.

2. **Layer Selection Strategy (Top-10 Sensitivity).** Instead of replacing all routers, we apply VMoER selectively to the **10 most sensitive layers**. These layers were identified via the brittleness analysis in Appendix B.1 (Figure 9b), corresponding to the depths where routing decisions were most fragile to perturbation. All other layers remain standard deterministic MoE layers.

3. **Stage 2: Variational Inference Learning (10 Epochs).** We **freeze** all model parameters—including the pre-trained router weights $\mathbf{W}_r$ and the LoRA adapters learned in Stage 1. We then initialise and optimise *only* the variational inference networks $\phi$ on the selected 10 layers for 10 epochs. This isolation ensures that any improvement in calibration is strictly attributable to the probabilistic routing in the critical layers.

**Hyperparameters & Optimisation**  All models were trained using the **AdamW** optimiser. To evaluate the robustness and transferability of our method, we performed hyperparameter grid search **exclusively on the Granite-MoE-3B model**. The optimal settings identified on Granite were then transferred directly to Qwen-MoE and DeepSeek-MoE without further tuning.

The final settings used for the reported results are:

- **Learning Rate:** $1 \times 10^{-4}$ with a cosine decay schedule (warmup ratio 0.05).

- **Batch Size:** Variable per model size (2, 4, or 8) with gradient accumulation steps adjusted to maintain a consistent **effective batch size of 16**.

- **Epochs:** 10 (with early stopping based on validation NLL).

- **KL Weight $\beta$:** Selected per dataset (on Granite), typically $\beta \in \{0.01, 0.1\}$ for VGLR and $\beta \in \{0.05, 0.1\}$ for VTSR to balance the trade-off between calibration (entropy) and accuracy.

- **Monte Carlo Samples:** $S = 1$ during training (reparameterisation), $S = 35$ during evaluation (to estimate ECE/NLL robustly).

**Compute Resources** Experiments were conducted on NVIDIA A100 (80GB) GPUs. Due to the lightweight nature of the inference networks (adding $< 1\%$ parameters) and the frozen backbone, training typically converged within 1–3 hours per dataset.

**Full Quantitative Results** Table 7 presents the comprehensive performance metrics across four In-Distribution datasets. We report **Accuracy (ACC ↑)** to measure task performance, and **Expected Calibration Error (ECE ↓)**, **Negative Log-Likelihood (NLL ↓)** and **Maximum Calibration Error (MCE ↓)** to measure uncertainty quality.

*Table 7.* **Main Results: Calibration & Performance by Dataset.** We compare standard deterministic routing against weight-space and our proposed VMoER across three MoE architectures. Results are reported as Mean$_{(std)}$ over 5 runs. Our methods (VGLR, VTSR) consistently reduce calibration error without degrading accuracy.

| Method | | Granite-MoE | | | | Qwen-MoE | | | | DeepSeek-MoE | | | |
|---|---|---|---|---|---|---|---|---|---|---|---|---|---|
| | | ACC ↑ | NLL ↓ | ECE ↓ | MCE ↓ | ACC ↑ | NLL ↓ | ECE ↓ | MCE ↓ | ACC ↑ | NLL ↓ | ECE ↓ | MCE ↓ |
| *Dataset: ARC-Challenge (Reasoning)* | | | | | | | | | | | | | |
| Base | MAP | **0.882** | 0.923 | 0.201 | 0.428 | 0.758 | 1.852 | 0.252 | 0.624 | 0.580 | 1.462 | 0.237 | 0.826 |
| | Temp-Scale | 0.824$_{(.004)}$ | 0.208$_{(.006)}$ | 0.038$_{(.007)}$ | 0.284$_{(.003)}$ | **0.762**$_{(.003)}$ | 1.422$_{(.001)}$ | 0.102$_{(.001)}$ | 0.304$_{(.005)}$ | **0.592**$_{(.003)}$ | 1.242$_{(.008)}$ | 0.188$_{(.006)}$ | 0.646$_{(.003)}$ |
| Weight | MCDR | 0.880$_{(.003)}$ | 0.146$_{(.006)}$ | 0.028$_{(.003)}$ | 0.228$_{(.007)}$ | 0.732$_{(.003)}$ | 1.134$_{(.001)}$ | 0.198$_{(.011)}$ | 0.446$_{(.011)}$ | 0.564$_{(.005)}$ | 1.084$_{(.009)}$ | 0.192$_{(.023)}$ | 0.848$_{(.003)}$ |
| | SWAG | 0.872$_{(.003)}$ | 0.138$_{(.006)}$ | 0.030$_{(.007)}$ | 0.266$_{(.002)}$ | 0.746$_{(.002)}$ | 1.122$_{(.006)}$ | 0.202$_{(.010)}$ | 0.387$_{(.002)}$ | 0.568$_{(.002)}$ | 1.183$_{(.001)}$ | 0.198$_{(.004)}$ | 0.842$_{(.009)}$ |
| Ours | VGLR-MF | 0.878$_{(.004)}$ | 0.125$_{(.005)}$ | 0.016$_{(.002)}$ | 0.196$_{(.002)}$ | 0.748$_{(.007)}$ | 0.902$_{(.002)}$ | 0.092$_{(.009)}$ | 0.252$_{(.006)}$ | 0.562$_{(.001)}$ | 0.724$_{(.004)}$ | 0.158$_{(.006)}$ | 0.562$_{(.012)}$ |
| | VGLR-FC | 0.880$_{(.006)}$ | **0.122**$_{(.001)}$ | **0.012**$_{(.006)}$ | **0.185**$_{(.003)}$ | 0.748$_{(.002)}$ | **0.892**$_{(.007)}$ | **0.022**$_{(.012)}$ | **0.204**$_{(.009)}$ | 0.572$_{(.004)}$ | **0.422**$_{(.001)}$ | **0.146**$_{(.002)}$ | 0.563$_{(.004)}$ |
| | VTSR | 0.872$_{(.002)}$ | 0.164$_{(.014)}$ | 0.020$_{(.004)}$ | 0.208$_{(.018)}$ | 0.760$_{(.009)}$ | 0.924$_{(.016)}$ | 0.038$_{(.009)}$ | 0.286$_{(.022)}$ | 0.584$_{(.009)}$ | 0.672$_{(.003)}$ | 0.178$_{(.003)}$ | **0.542**$_{(.009)}$ |
| *Dataset: OpenBookQA (Common Sense)* | | | | | | | | | | | | | |
| Base | MAP | **0.746** | 1.380 | 0.252 | 0.472 | 0.804 | 1.370 | 0.127 | 0.273 | 0.802 | 1.120 | 0.168 | 0.321 |
| | Temp-Scale | 0.716$_{(.005)}$ | 0.773$_{(.049)}$ | 0.107$_{(.009)}$ | 0.201$_{(.013)}$ | 0.818$_{(.011)}$ | 1.120$_{(.052)}$ | 0.102$_{(.016)}$ | 0.224$_{(.020)}$ | **0.808**$_{(.012)}$ | 0.870$_{(.057)}$ | 0.146$_{(.022)}$ | 0.402$_{(.017)}$ |
| Weight | MCDR | 0.734$_{(.002)}$ | **0.650**$_{(.022)}$ | 0.037$_{(.028)}$ | 0.298$_{(.008)}$ | 0.798$_{(.009)}$ | 0.982$_{(.018)}$ | 0.064$_{(.023)}$ | 0.152$_{(.003)}$ | 0.794$_{(.015)}$ | 0.729$_{(.014)}$ | 0.103$_{(.023)}$ | 0.302$_{(.002)}$ |
| | SWAG | 0.736$_{(.002)}$ | 0.652$_{(.030)}$ | 0.041$_{(.013)}$ | 0.290$_{(.007)}$ | 0.782$_{(.006)}$ | 0.912$_{(.031)}$ | 0.098$_{(.005)}$ | 0.146$_{(.009)}$ | 0.786$_{(.014)}$ | 0.665$_{(.026)}$ | 0.140$_{(.002)}$ | 0.288$_{(.008)}$ |
| Ours | VGLR-MF | 0.742$_{(.001)}$ | 0.654$_{(.019)}$ | 0.026$_{(.009)}$ | 0.293$_{(.004)}$ | 0.796$_{(.003)}$ | 0.724$_{(.025)}$ | 0.028$_{(.007)}$ | 0.124$_{(.012)}$ | 0.788$_{(.011)}$ | 0.476$_{(.019)}$ | 0.067$_{(.009)}$ | 0.174$_{(.008)}$ |
| | VGLR-FC | 0.740$_{(.001)}$ | 0.652$_{(.021)}$ | **0.015**$_{(.008)}$ | **0.152**$_{(.004)}$ | 0.802$_{(.003)}$ | 0.694$_{(.014)}$ | **0.014**$_{(.008)}$ | **0.098**$_{(.003)}$ | 0.800$_{(.004)}$ | 0.447$_{(.018)}$ | **0.054**$_{(.015)}$ | **0.146**$_{(.005)}$ |
| | VTSR | 0.736$_{(.003)}$ | 0.667$_{(.025)}$ | 0.052$_{(.023)}$ | 0.293$_{(.014)}$ | **0.818**$_{(.003)}$ | **0.682**$_{(.032)}$ | 0.022$_{(.028)}$ | 0.106$_{(.008)}$ | 0.798$_{(.004)}$ | **0.431**$_{(.031)}$ | 0.060$_{(.032)}$ | 0.262$_{(.008)}$ |
| *Dataset: SciQ (Science Knowledge)* | | | | | | | | | | | | | |
| Base | MAP | 0.850 | 0.791 | 0.223 | 0.452 | 0.872 | 1.320 | 0.274 | 0.648 | 0.832 | 1.124 | 0.279 | 0.898 |
| | Temp-Scale | 0.878$_{(.002)}$ | 0.309$_{(.002)}$ | 0.047$_{(.003)}$ | 0.649$_{(.005)}$ | **0.876**$_{(.002)}$ | 1.123$_{(.009)}$ | 0.111$_{(.004)}$ | 0.669$_{(.002)}$ | 0.828$_{(.017)}$ | 0.642$_{(.007)}$ | 0.121$_{(.002)}$ | 0.511$_{(.003)}$ |
| Weight | MCDR | 0.880$_{(.006)}$ | 0.296$_{(.003)}$ | 0.029$_{(.006)}$ | 0.366$_{(.007)}$ | 0.854$_{(.002)}$ | 0.884$_{(.009)}$ | 0.199$_{(.000)}$ | 0.586$_{(.015)}$ | 0.826$_{(.002)}$ | 0.296$_{(.002)}$ | 0.204$_{(.002)}$ | 0.648$_{(.002)}$ |
| | SWAG | 0.879$_{(.001)}$ | **0.291**$_{(.004)}$ | 0.031$_{(.004)}$ | 0.392$_{(.002)}$ | 0.829$_{(.001)}$ | 0.875$_{(.007)}$ | 0.203$_{(.012)}$ | 0.511$_{(.008)}$ | 0.828$_{(.003)}$ | 0.298$_{(.007)}$ | 0.212$_{(.006)}$ | 0.592$_{(.009)}$ |
| Ours | VGLR-MF | **0.884**$_{(.004)}$ | 0.297$_{(.004)}$ | 0.019$_{(.002)}$ | 0.387$_{(.002)}$ | 0.862$_{(.003)}$ | 0.674$_{(.010)}$ | 0.095$_{(.000)}$ | 0.443$_{(.006)}$ | 0.838$_{(.007)}$ | 0.297$_{(.010)}$ | 0.095$_{(.010)}$ | 0.484$_{(.008)}$ |
| | VGLR-FC | **0.884**$_{(.005)}$ | 0.298$_{(.005)}$ | **0.013**$_{(.002)}$ | **0.320**$_{(.005)}$ | 0.858$_{(.002)}$ | 0.668$_{(.004)}$ | **0.023**$_{(.005)}$ | **0.339**$_{(.002)}$ | **0.840**$_{(.001)}$ | 0.292$_{(.004)}$ | **0.016**$_{(.001)}$ | **0.220**$_{(.005)}$ |
| | VTSR | 0.874$_{(.002)}$ | 0.299$_{(.002)}$ | 0.022$_{(.002)}$ | 0.352$_{(.002)}$ | 0.870$_{(.005)}$ | **0.659**$_{(.013)}$ | 0.040$_{(.001)}$ | 0.430$_{(.010)}$ | 0.830$_{(.009)}$ | **0.291**$_{(.004)}$ | 0.047$_{(.002)}$ | 0.360$_{(.006)}$ |
| *Dataset: MedMCQA (Medical)* | | | | | | | | | | | | | |
| Base | MAP | **0.550** | 1.290 | 0.183 | 0.288 | **0.542** | 1.285 | 0.190 | 0.286 | **0.582** | 1.294 | 0.212 | 1.462 |
| | Temp-Scale | 0.486$_{(.004)}$ | **1.170**$_{(.003)}$ | 0.039$_{(.005)}$ | 0.097$_{(.005)}$ | 0.491$_{(.007)}$ | 1.179$_{(.006)}$ | 0.122$_{(.008)}$ | 0.193$_{(.001)}$ | 0.542$_{(.004)}$ | 1.182$_{(.004)}$ | 0.124$_{(.009)}$ | 0.282$_{(.004)}$ |
| Weight | MCDR | 0.494$_{(.005)}$ | 1.170$_{(.005)}$ | 0.050$_{(.003)}$ | **0.096**$_{(.008)}$ | 0.482$_{(.002)}$ | 1.174$_{(.001)}$ | 0.125$_{(.001)}$ | **0.097**$_{(.005)}$ | 0.518$_{(.005)}$ | 1.175$_{(.004)}$ | 0.096$_{(.003)}$ | 0.512$_{(.004)}$ |
| | SWAG | 0.486$_{(.005)}$ | 1.200$_{(.006)}$ | 0.096$_{(.005)}$ | 0.179$_{(.004)}$ | 0.490$_{(.007)}$ | 1.212$_{(.001)}$ | 0.188$_{(.010)}$ | 0.280$_{(.012)}$ | 0.526$_{(.004)}$ | 1.220$_{(.005)}$ | 0.068$_{(.001)}$ | 0.458$_{(.002)}$ |
| Ours | VGLR-MF | 0.492$_{(.002)}$ | 1.170$_{(.001)}$ | 0.039$_{(.001)}$ | 0.103$_{(.002)}$ | 0.497$_{(.003)}$ | 1.178$_{(.003)}$ | 0.097$_{(.000)}$ | 0.145$_{(.000)}$ | 0.554$_{(.003)}$ | 1.185$_{(.006)}$ | 0.056$_{(.001)}$ | 0.342$_{(.005)}$ |
| | VGLR-FC | 0.494$_{(.004)}$ | 1.170$_{(.004)}$ | **0.022**$_{(.003)}$ | 0.108$_{(.007)}$ | 0.490$_{(.007)}$ | 1.175$_{(.003)}$ | **0.030**$_{(.001)}$ | 0.102$_{(.013)}$ | 0.562$_{(.006)}$ | **1.146**$_{(.008)}$ | **0.038**$_{(.001)}$ | **0.124**$_{(.007)}$ |
| | VTSR | 0.476$_{(.005)}$ | 1.170$_{(.002)}$ | 0.053$_{(.005)}$ | 0.113$_{(.008)}$ | 0.479$_{(.009)}$ | **1.167**$_{(.014)}$ | 0.051$_{(.002)}$ | 0.103$_{(.002)}$ | 0.540$_{(.003)}$ | 1.192$_{(.003)}$ | 0.042$_{(.007)}$ | 0.456$_{(.012)}$ |

## D.3. Experiment II: Out-of-Distribution Detection & Uncertainty Analysis

In this experiment, we assess the ability of the learned variational routers to detect distributional shifts. We use the **OBQA** dataset as the In-Distribution (ID) source and evaluate on the Near-OoD (**ARC**) and Far-OoD (**SciQ, MMLU, MedMCQA**) datasets defined in Table 6.

**Uncertainty Signals** To quantify the router's epistemic uncertainty, we evaluate four distinct signals. We denote the input token as **u** and the number of experts as $N$.

1. **Gate Entropy (Gate-Ent):** The standard baseline metric, defined as the Shannon entropy of the final expert selection probabilities. High entropy implies the router is "confused" between experts.

$$\mathcal{U}_{\text{Ent}}(\mathbf{u}) = -\sum_{i=1}^{N} p_i \log p_i \tag{33}$$

For **VTSR**, this metric naturally adapts to the learned uncertainty by computing the entropy of the temperature-scaled probabilities: $\mathcal{H}(\text{softmax}(\mathbf{l}_{\text{det}}/T_\phi(\mathbf{u})))$.

2. **Inferred Logit Variance (Inf-Logit-Var):** Specific to **VGLR-FC**, this signal leverages the analytically predicted covariance. It is calculated as the trace of the posterior covariance matrix $\boldsymbol{\Sigma}_{\text{post}} = \mathbf{L}\mathbf{L}^\top$, representing the total magnitude of uncertainty in the logit space.

$$\mathcal{U}_{\text{VGLR}}(\mathbf{u}) = \text{tr}(\boldsymbol{\Sigma}_{\text{post}}) = \|\mathbf{L}\|_F^2 = \sum_{i,j} L_{ij}^2 \tag{34}$$

3. **Inferred Temperature (Inf-Temp):** Specific to **VTSR**, this signal is the raw scalar temperature predicted by the inference network. A higher temperature explicitly indicates that the model desires a flatter (more uniform) distribution to account for ambiguity.

$$\mathcal{U}_{\text{VTSR}}(\mathbf{u}) = T_\phi(\mathbf{u}) \tag{35}$$

4. **Monte Carlo Logit Variance (MC-Logit-Var):** A general metric for any stochastic router (including VGLR at inference). We perform $S$ stochastic forward passes to generate a set of logit vectors $\{\mathbf{l}^{(s)}\}_{s=1}^S$. The signal is the total variance of the logit vector norm:

$$\mathcal{U}_{\text{MC}}(\mathbf{u}) = \frac{1}{S-1} \sum_{s=1}^S \|\mathbf{l}^{(s)} - \bar{\mathbf{l}}\|_2^2 \tag{36}$$

**Full Quantitative Results**    Table 8 details the shift detection performance. We report the **Area Under the ROC Curve (AUROC ↑)** and **Area Under the Precision-Recall Curve (AUPRC ↑)**, evaluating how well each uncertainty signal distinguishes between ID (OBQA) and OoD tokens.

*Table 8.* **OoD Detection Performance across Routing Methods and Signals.** We compare the effectiveness of uncertainty signals for Out-of-Distribution detection. **Bold** indicates the best performance across all signals for a given model. *Inf-Logit-Var* consistently performs best for VGLR methods, while VTSR often benefits from *Gate-Ent*. Both performs equally best overall.

| | | | Near-Domain Shift | | | | Far-Domain Shift | | | | Average | |
| | | | OBQA → ARC-E | | OBQA → ARC-C | | OBQA → MedMCQA | | OBQA → MMLU-Law | | Overall | |
| Model | Method | Signal | AUROC | AUPRC | AUROC | AUPRC | AUROC | AUPRC | AUROC | AUPRC | AUROC | AUPRC |
|---|---|---|---|---|---|---|---|---|---|---|---|---|
| Granite | MAP | Gate-Ent | 0.612 | 0.596 | 0.633 | 0.626 | 0.679 | 0.645 | 0.683 | 0.686 | 0.652 | 0.638 |
| | MCDR | Gate-Ent | 0.612 | 0.599 | 0.632 | 0.610 | 0.684 | 0.651 | 0.691 | 0.672 | 0.655 | 0.633 |
| | | MC-Logit-Var | 0.610 | 0.583 | 0.677 | 0.623 | 0.786 | 0.723 | 0.793 | 0.765 | 0.717 | 0.674 |
| | VGLR-MF | Gate-Ent | **0.622** | 0.603 | 0.642 | 0.622 | 0.682 | 0.637 | 0.673 | 0.664 | 0.655 | 0.632 |
| | | Inf-Logit-Var | 0.617 | 0.587 | 0.672 | 0.669 | 0.835 | 0.793 | 0.824 | 0.763 | 0.737 | 0.703 |
| | VGLR-FC | Gate-Ent | 0.615 | 0.605 | 0.652 | 0.632 | 0.692 | 0.642 | 0.677 | 0.674 | 0.659 | 0.638 |
| | | Inf-Logit-Var | 0.609 | 0.578 | **0.709** | 0.665 | **0.844** | 0.773 | **0.834** | **0.810** | **0.749** | 0.707 |
| | VTSR | Gate-Ent | 0.621 | **0.624** | 0.693 | **0.672** | 0.834 | **0.813** | 0.824 | 0.802 | 0.743 | **0.728** |
| | | Inf-Temp | 0.502 | 0.501 | 0.498 | 0.503 | 0.512 | 0.492 | 0.523 | 0.502 | 0.509 | 0.500 |
| Qwen | MAP | Gate-Ent | 0.633 | 0.621 | 0.649 | 0.632 | 0.704 | 0.678 | 0.721 | 0.708 | 0.677 | 0.660 |
| | MCDR | Gate-Ent | 0.632 | 0.625 | 0.651 | 0.641 | 0.718 | 0.686 | 0.720 | 0.688 | 0.680 | 0.660 |
| | | MC-Logit-Var | 0.637 | 0.606 | 0.701 | 0.638 | 0.813 | 0.758 | 0.816 | 0.791 | 0.742 | 0.698 |
| | VGLR-MF | Gate-Ent | 0.641 | 0.638 | 0.661 | 0.654 | 0.714 | 0.670 | 0.705 | 0.687 | 0.680 | 0.662 |
| | | Inf-Logit-Var | 0.626 | 0.619 | 0.699 | 0.673 | 0.849 | 0.802 | 0.849 | 0.786 | 0.756 | 0.720 |
| | VGLR-FC | Gate-Ent | 0.634 | 0.622 | 0.681 | 0.654 | 0.721 | 0.660 | 0.712 | 0.708 | 0.687 | 0.661 |
| | | Inf-Logit-Var | 0.631 | 0.598 | **0.729** | 0.682 | 0.864 | 0.820 | **0.869** | **0.831** | **0.773** | 0.733 |
| | VTSR | Gate-Ent | **0.642** | **0.647** | 0.726 | **0.696** | **0.869** | **0.846** | 0.854 | 0.818 | **0.773** | **0.752** |
| | | Inf-Temp | 0.537 | 0.516 | 0.515 | 0.521 | 0.530 | 0.523 | 0.550 | 0.532 | 0.533 | 0.523 |
| DeepSeek | MAP | Gate-Ent | 0.597 | 0.586 | 0.613 | 0.616 | 0.665 | 0.625 | 0.671 | 0.669 | 0.637 | 0.624 |
| | MCDR | Gate-Ent | 0.602 | 0.585 | 0.622 | 0.599 | 0.670 | 0.638 | 0.672 | 0.661 | 0.642 | 0.621 |
| | | MC-Logit-Var | 0.594 | 0.568 | 0.660 | 0.611 | 0.766 | 0.704 | 0.778 | 0.755 | 0.700 | 0.660 |
| | VGLR-MF | Gate-Ent | 0.611 | 0.586 | 0.625 | 0.609 | 0.666 | 0.624 | 0.657 | 0.653 | 0.640 | 0.618 |
| | | Inf-Logit-Var | 0.608 | 0.571 | 0.655 | 0.649 | 0.821 | 0.774 | 0.807 | 0.749 | 0.723 | 0.686 |
| | VGLR-FC | Gate-Ent | **0.615** | 0.590 | 0.635 | 0.612 | 0.677 | 0.622 | 0.657 | 0.656 | 0.646 | 0.620 |
| | | Inf-Logit-Var | 0.589 | 0.566 | **0.690** | 0.652 | **0.828** | 0.755 | **0.823** | **0.795** | **0.733** | 0.692 |
| | VTSR | Gate-Ent | 0.606 | **0.609** | 0.677 | **0.657** | 0.819 | **0.793** | 0.814 | 0.792 | 0.729 | **0.713** |
| | | Inf-Temp | 0.521 | 0.517 | 0.508 | 0.506 | 0.613 | 0.582 | 0.595 | 0.527 | 0.559 | 0.533 |

## D.4. Experiment III: Stability Verification

In this final experiment, we verify whether the variational routers mitigate the structural brittleness identified in our motivational analysis (Appendix B.1). We subject the trained VMoER models to the same perturbation protocol used to diagnose the baseline's instability.

**Methodology**   We follow the exact setup described in Appendix B.1.

- **Perturbation:** We inject Gaussian noise $\epsilon \sim \mathcal{N}(\mathbf{0}, \sigma^2 \mathbf{I})$ into the input embeddings of the 10 selected MoE layers, with noise levels $\sigma \in \{0.001, 0.002, 0.005, 0.010\}$.

- **Metric:** We measure the **Jaccard Similarity** $J(E_{\text{orig}}, E_{\text{pert}})$ between the sets of experts selected for the clean input versus the perturbed input. A higher Jaccard score indicates greater robustness.

**Full Quantitative Results**   Table 9 compares the stability of the Baseline (Deterministic) router against our Variational variants. For the probabilistic routers (VGLR/VTSR), we report average JS among all modified layers.

*Table 9.* **Routing Stability under Input Perturbation.** We report the average Jaccard Similarity of expert selection across the *modified probabilistic layers* for each model.

| | Granite | | | | DeepSeek | | | | QWen | | | |
|---|---|---|---|---|---|---|---|---|---|---|---|---|
| Noise ($\sigma$) | MAP | MCDR | VGLR-FC | VTSR | MAP | MCDR | VGLR-FC | VTSR | MAP | MCDR | VGLR-FC | VTSR |
| 0.001 | 0.946 | 0.954 | **0.970** | 0.962 | 0.956 | **0.958** | 0.948 | 0.954 | 0.938 | 0.944 | **0.956** | 0.948 |
| 0.002 | 0.922 | 0.932 | **0.940** | 0.934 | 0.918 | **0.944** | 0.940 | 0.922 | 0.924 | 0.932 | **0.946** | 0.936 |
| 0.005 | 0.650 | 0.822 | **0.886** | 0.840 | 0.712 | 0.732 | **0.758** | 0.746 | 0.614 | 0.662 | **0.672** | 0.668 |
| 0.010 | 0.422 | 0.654 | **0.702** | 0.692 | 0.624 | 0.684 | **0.718** | 0.702 | 0.532 | 0.582 | 0.612 | **0.614** |

# E. Efficiency Analysis Details

We provide the derivations for the memory and computational overhead analysis presented in Section 5.5. We assume a **Parallel Execution** environment, where all $S$ Monte Carlo samples are processed in a single batch to minimize wall-clock latency.

**Measurement Protocol**  Theoretical complexity is derived using standard Big-O notation. Empirical measurements (Actual Add. GFLOPs reported in the main text) were conducted using the `fvcore.nn.FlopCountAnalysis` library (Research, 2019) on a single forward pass of the Granite-3B-MoE model.

**Notation**

- $L$: Number of modified MoE layers (10).
- $N$: Experts per layer (40).
- $D$: Hidden dimension (1536).
- $S$: Monte Carlo samples (35).
- $H$: Inference network dimension ($D/4 = 384$).

## E.1. Activation Memory Overhead

The critical constraint is the peak memory required to hold parameters during the forward pass.

**Weight-Space Baselines (MCDR, SWAGR)**  Typically, MC-Dropout (MCDR) is considered memory-efficient because it requires no extra parameters. However, in a **latency-critical** setting, running $S$ sequential passes is unacceptable. To parallelise MCDR, we must replicate the masked weight matrices $S$ times in memory (or store $S$ distinct large masks). Thus, both MCDR and SWAGR scale linearly with $S$:

$$\mathcal{M}_{\text{Weight}} = L \times (S - 1) \times (D \cdot N) \tag{37}$$

For $S = 35$, this results in storing 34 additional copies of the dense router weights, causing the high overhead ($\sim$2.6%) observed in Table 4.

**VMoER Methods**  Our methods rely on fixed inference networks. The memory cost is constant regardless of $S$, as samples are transient vectors, not matrices.

- **MFVR:** Backbone ($DH$) + Mean/Var Heads ($2HN$).

- **VGLR-FC:** Backbone ($DH$) + Mean Head ($HN$) + Cholesky Head ($H \cdot \frac{N(N+1)}{2} \approx \frac{1}{2} HN^2$).

$$\mathcal{M}_{\text{FC}} \approx L \cdot H \cdot (D + N + 0.5N^2) \tag{38}$$

- **VTSR:** Backbone ($DH$) + Scalar Head ($H$).

### E.2. Computational Overhead (FLOPs)

We assume a standard linear layer requires $2 \cdot \text{in} \cdot \text{out}$ FLOPs.

**Weight-Space Methods**    Every sample requires a full pass through the router's projection layer ($D \rightarrow N$).

$$\mathcal{C}_{\text{Weight}} = L \times S \times (2DN) \tag{39}$$

**VMoER Methods**    The heavy feature extraction ($D \rightarrow H$) runs only once per token ($S = 1$). The scaling with $S$ affects only the cheap sampling step.

- **VGLR-MF:** Double-head one-hidden-layer MLP adds approximately $2DH + 4HN$ FLOPs. Reparameterisation trick for $S$ samples adds $S \times 2N$ FLOPs. Total overhead is the sum of these two.

- **VGLR-FC:** The most expensive variant involves generating the Cholesky factor ($H \rightarrow N^2/2$) and sampling ($S$ matrix-vector products).

$$\mathcal{C}_{\text{VGLR-FC}} = L \left( \underbrace{2DH + HN^2}_{\text{Network } \phi} + \underbrace{2SN^2}_{\text{Sampling}} \right) \tag{40}$$

Even with the quadratic term $N^2$, since $N \ll D$ (40 vs 1536), the total FLOPs remain roughly $2\times$ lower than the weight-space baseline.

- **VTSR:** The temperature prediction network adds approximately $2DH + 2H$ FLOPs (Backbone + Scalar Head). This is followed by $N$ divisions to scale the logits.

$$\mathcal{C}_{\text{VTSR}} = L \times (2DH + 2H + N) \tag{41}$$

Note that this analysis does not include the cost of averaging multiple post-softmax outputs. If considered strictly from a theoretical standpoint, VTSR is even more efficient as it does not require Monte Carlo sampling ($S = 1$) during the forward pass logic itself.

## F. Ablation Studies

In this section, we isolate two critical design choices: the quantity of modified layers ($L$) and their placement strategy. All ablations are performed on the **Granite-MoE-3B** model. Specifically, we use the **VGLR-FC** method to analyse the sensitivity to $L$, while the strategy comparison evaluates representative methods from each family (**MCDR**, **VGLR-FC**, and **VTSR**).

**Ablation A: Sensitivity to Number of Layers** ($L$)    We first determine the optimal number of layers to modify. We vary $L \in \{1, 2, 4, 8, 10, 12, 15\}$, selecting layers based on the sensitivity ranking from Appendix B.1. Figure 12 (Left) illustrates the trade-off: calibration performance (ECE) improves steadily up to $L = 10$, after which marginal gains diminish while variance increases.

**Ablation B: Layer Selection Strategy**    We then fix $L = 10$ and evaluate *where* these layers should be placed. We compare our adaptive **Susceptible-10** strategy against three fixed baselines:

- **First-10:** Modifying the earliest layers (0–9).

- **Middle-10:** Modifying the intermediate layers (11–20).

- **Last-10:** Modifying the final layers (22–31).

Table 10 confirms that applying variance to the most brittle (susceptible) layers consistently yields the best results across all primary dimensions: **ECE** for calibration (OBQA), **AUROC** for uncertainty quantification (OBQA $\rightarrow$ MMLU-Law), and **Jaccard Similarity Improvment** ($\Delta$**JS**) for Increased stability against noise ($\sigma = 0.005$).

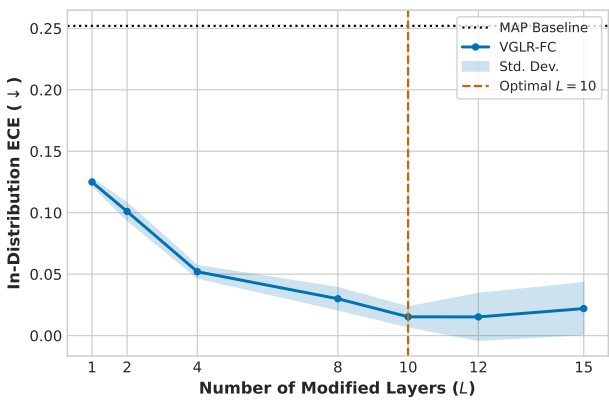

*Figure 12.* **Sensitivity Analysis ($L$).** Effect of increasing the number of variational layers on In-Distribution Calibration (ECE). $L = 10$ represents the optimal trade-off point.

*Table 10.* **Layer Selection Strategy.** Comparison of adaptive vs. fixed placement ($L = 10$). The *Susceptible* strategy (Ours) targets the specific layers identified as brittle.

| Method | Strategy | $\triangle$JS $\uparrow$ | ECE $\downarrow$ | AUROC $\uparrow$ |
|---|---|---|---|---|
| **MCDR** | First-10 | 0.022 | 0.302 | 0.638 |
| | Middle-10 | 0.062 | 0.163 | 0.674 |
| | Last-10 | 0.092 | 0.128 | 0.703 |
| | **Susceptible** | **0.172** | **0.037** | **0.793** |
| **VGLR** | First-10 | 0.018 | 0.242 | 0.698 |
| | Middle-10 | 0.098 | 0.193 | 0.736 |
| | Last-10 | 0.156 | 0.104 | 0.792 |
| | **Susceptible** | **0.216** | **0.015** | **0.834** |
| **VTSR** | First-10 | 0.003 | 0.247 | 0.703 |
| | Middle-10 | 0.076 | 0.198 | 0.791 |
| | Last-10 | 0.112 | 0.145 | 0.807 |
| | **Susceptible** | **0.190** | **0.052** | **0.824** |

## G. MoEs Dominance of Current LMArena

At the time of writing 18 of the top 50 models on LMArena (Chiang et al., 2024) are non-proprietary. These 18 models stem from five families of models each of which are MoE architectures: Kimi (Team et al., 2025), GLM (Zeng et al., 2025), DeepSeek (DeepSeek-AI et al., 2025), Qwen (Qwen, 2024), and Mistral (Mistral AI, 2025).

## H. Additional Related Works

In this section, we provide coverage of additional related works beyond the core discussion in the main text.

**Bayesian Deep Learning.** A Bayesian approach to learning in deep neural networks has been investigated for more than three decades (Neal, 2012). Traditionally, a Bayesian perspective is applied to the entire learnable parameter space of a deep neural network; while exact inference is intractable, many approximate inference strategies have been investigated including HMC (Neal et al., 2011), Bayes by Backprop (BBB) (Blundell et al., 2015), Stochastic Weight Averaging - Gaussian (SWAG) (Maddox et al., 2019), NoisyAdam (NA) (Zhang et al., 2018), and Variational Online Gauss-Newton (VOGN) (Osawa et al., 2019). The quality of the inference can have substantial effects on the resulting predictive uncertainty. For example, while it has been shown that Monte Carlo Dropout (MCDropout) (Gal & Ghahramani, 2016) can be viewed as an approximation of a Bayesian posterior, improvements in methodology aimed at more faithful inference can substantially improve predictive uncertainty (Li & Gal, 2017).

**Uncertainty and Robustness.** There is a long line of literature connecting the notion of Bayesian inference and robustness. In adversarial robustness, the kind that provides the motivating example for this paper it has been established that exact inference should have positive effects on a neural networks robustness (Gilmer et al., 2018; Bekasov & Murray, 2018). This general notion was strengthened in the more specific setting of gradient-based adversarial attacks (Carbone et al., 2020; Bortolussi et al., 2024). Standard advances in adversarial robustness for deterministic neural networks have also been leveraged to improve approximate Bayesian inference (Liu et al., 2018; Wicker et al., 2021) and formal certificates of robustness have also been investigated (Cardelli et al., 2019; Wicker et al., 2020). Beyond input perturbations, Bayes has also been linked to robustness of the optimization process (Möllenhoff & Khan, 2022; Nickl et al., 2023). Provided the numerical issues observed with training MoE architectures, this may serve as another valuable future direction of investigation for future works in Bayesian transformers.

