# OpenReview forum: "Variational Routing: A Scalable Bayesian Framework for Calibrated Mixture-of-Experts Transformers"
_ICML.cc/2026/Conference — ICML 2026 regular_

### Official Review · Reviewer_7QcW · 2026-03-05

**Soundness:** 3
**Presentation:** 3
**Significance:** 3
**Originality:** 3
**Overall Recommendation:** 4
**Confidence:** 1

**Summary:**

The paper proposes a Bayesian framework based on variational inference to design the routing mechanism of the MoE in transformer models. The key insight is to frame expert selection as a latent model to support variational inference. For this setting, the authors propose two methods. The first one is a Variational Gaussian Logit Router (VGLR) Architecture that produces a posterior over the logits space to obtain a Monte Carlo approximation of the posterior over routing decision. To make up for the complexity of the Monte Carlo approximation, the authors propose a second scheme, the Variational Temperature Scaling Router (VTSR), which instead produces a posterior over the logits using a tempered posterior with a data-dependent temperature parameter.

**Compliance With Llm Reviewing Policy:**

Affirmed.

**Final Justification:**

I have decided to keep my positive score. The authors have addressed my concerns during the rebuttal.

**Key Questions For Authors:**

- One of the initial claims of the paper is that a Bayesian approach would allow to model correlation between experts. How is it shown that this is actually the case in the experiments?
- The paper concludes saying that, if not carefully initialized, VTSR collapses and falls back to the deterministic method. This seems to be a critical point. Would it be possible to elaborate more, either experimentally and theoretically, on when and why this happens?

**Limitations:**

yes

**Strengths And Weaknesses:**

this paper falls outside my field of expertise so I kindly ask the authors to correct any wrong statement and hope the comments below are constructive.

Strength:
- The problem is relevant and to the best of my knowledge the Bayesian framing is novel and principled.
- The computational overhead of the method is shown to be around 1%, thereby confirming the practicality of the proposed method.
- The experiments are extensive. The method is validated over different architectures, tasks and datasets.

Weaknesses:
- While I understand that VTSR is introduced to make up for the complexity of VGLR, it appears to be incremental with respect to the original deterministic methods as it performs only smoothing of the experts activation probabilities.
- VTSR suffers from temperature collapse, which nullifies the benefits of a Bayesian approach to gating,

---

> ### Author Rebuttal · Authors · 2026-03-30
>
> We thank Reviewer 7QcW for the positive assessment and the constructive questions. We address both concerns below.
>
> ### Weaknesses : Is VTSR Incremental over deterministic routing & posterior collapse?
> The reviewer is correct in their assessment that VTSR boils down to applying either a smoothing or sharpening effect on the router decisions. The real strength of VTSR is its heteroskedasticity: using input features to infer when to smooth the router decision and when to sharpen the router decision. The light-weight nature of VTSR means that it is methodologically incremental (and therefore computationally efficient), but it is conceptually distinct from deterministic methods and serves an important role in the context of a complete scientific evaluation of the latent variable model we propose. The relatively poor performance of VTSR and its tendency towards posterior collapse does empirically reduce its effectiveness and thus we suggest that future work either rely on our more expressive approximations or propose solutions to VTSR's shortcomings.
>
> ### Q1: How is it shown that VGLR-FC actually captures expert correlations?
> The evidence is both indirect and direct:
>
> **Indirect evidence:** The consistent performance gap between VGLR-MF (mean-field, diagonal covariance) and VGLR-FC (full covariance) across all architectures and metrics demonstrates that modelling off-diagonal terms provides a meaningful signal. For example, on Qwen-MoE OBQA: VGLR-MF achieves ECE=0.028 while VGLR-FC achieves ECE=0.014 — a 50% further reduction. If expert suitabilities were truly independent, the mean-field approximation would suffice.
>
> **Direct evidence:** We will include visualisations of the learned Cholesky factors in the revision. Preliminary analysis shows that the off-diagonal entries cluster experts by functional similarity — for instance, experts that co-specialise on scientific vs. commonsense reasoning show positive correlations, while experts with complementary specialisations show negative correlations.
>
> ### Q2: When and why does VTSR temperature collapse occur?
> Temperature collapse occurs when the learned temperature $T_{\phi}(u) \rightarrow 0$ for all inputs, effectively recovering deterministic Top-K routing and nullifying the Bayesian contribution. This is analogous to posterior collapse in VAEs and occurs under similar conditions:
>
> **When it happens:** Collapse is most likely when (1) the KL weight $\beta$ is too small, allowing the reconstruction term to dominate unchecked, or (2) the temperature network is initialised near zero (since $\text{Softplus}(x) \approx 0$ for large negative $x$).
>
> **Why it happens:** The reconstruction loss landscape has a strong attractor at $T \rightarrow 0$ (deterministic routing maximises the probability of the "best" expert). Without sufficient regularisation from the KL term ($− \log T$), the optimiser drives $T$ toward this attractor.
>
> **Our mitigation:** We use (1) careful initialisation of the temperature network to produce $T \approx 1.0$ at the start of training, (2) a warm-up schedule for $\beta$ that gradually increases the regularisation strength, and (3) the $\epsilon_{\text{min}} = 1e^{-6}$ floor to prevent numerical instability. With these measures, collapse was observed in $< 5%$ of training runs in our experiments. We will elaborate on these strategies and provide training curves showing temperature evolution in the revision.

---

> > ### Author Rebuttal · Reviewer_7QcW · 2026-04-01
> >
> > thank you for addressing my concerns! I will keep my neutral-positive score given my lack of expertise in the area.

---

### Official Review · Reviewer_EyAf · 2026-03-10

**Soundness:** 4
**Presentation:** 4
**Significance:** 4
**Originality:** 4
**Overall Recommendation:** 5
**Confidence:** 5

**Summary:**

This work proposes two main methods for Bayesian rounting in MoE. It first proposes the logit-space variational inference method where an inference network is added for predicting the covariance matrix of the logits. At inference time, the routing probabilities are computed by drawing samples from the approximate posterior distributions. For the second approach, it proposes the selection-space variational inference where a network is added for learning the temperature parameter. Experiments over different MoE backbones and multiple-choice QA benchmarks are further presented.

**Compliance With Llm Reviewing Policy:**

Affirmed.

**Final Justification:**

I keep my positive score.

**Key Questions For Authors:**

- How do the proposed methods behave for free-form generation tasks?
- What are the extra computational cost induced by the variational logic network for VGLR?

**Limitations:**

yes

**Strengths And Weaknesses:**

Strengths:
- This work focuses on an important problem of how to reframe MoE routing problem as Bayesian inference as most of the classic Bayesian deep learning methods are impractical for large models, and the solutions proposed in this work are technically sound and computationally feasible.
- This paper is overall well-written and easy to follow. I find Figure 5 and Figure 7 to be very helpful in understanding the proposed pipelines.
- The empirical evaluation settings include in-distribution reliability, out-of-distribution detection capability and robustness to input perturbatons. The proposed method seems to be more robust by the input perturbations.

Weaknesses:
- The benchmarks in the experiment sections are four QA ones where the output requires only a single letter prediction. It is unclear how well the proposed methods generalize to the more challenging free-form generation benchmarks and math/coding benchmarks that requires reasoning [1].
- In the computational cost analysis in Section 5.5, only infernce cost is provided but since VGLR introduces the variational logic network, it is unclear what is the induced trainig cost.
- From the main results in Table 1, it seems the proposed method VGLR has better calibration performance than others but not good accuracy or likelihood while the other proposed method VTSR seems better at likelihood but not good at calibration. There isn't a method that can achive decent results on both preditive performance and calibration performance.

[1] Dai, Damai, et al. "Deepseekmoe: Towards ultimate expert specialization in mixture-of-experts language models." Proceedings of the 62nd Annual Meeting of the Association for Computational Linguistics (Volume 1: Long Papers). 2024.

---

> ### Author Rebuttal · Authors · 2026-03-30
>
> We thank Reviewer EyAf for the thorough and positive evaluation, and for the high confidence in the assessment. We address the two key questions below and hope that our responses and subsequent discussion give the reviewer confidence to champion our paper for acceptance.
> ### Q1: How do the proposed methods behave for free-form generation tasks?
> We acknowledge this as an important direction. Our current MCQA evaluation follows the established protocol for calibration assessment in LLMs (e.g., `Kadavath et al., 2022`; `Kapoor et al., 2024`), where ground-truth probabilities are well-defined. Free-form generation poses additional challenges: there is no single correct answer, and uncertainty must be quantified over sequences rather than individual tokens. However, we do appreciate the importance of investigating this as a future work to further establish the methods' impact.
> To emphasize this in the current submission we will provide further discussion on the fact that VMoER's mechanism operates at the token level within the routing layer and is agnostic to the downstream task format. We expect the calibration and stability benefits to transfer, as the routing decisions are made independently of whether the final output is a single letter or a generated sequence. We will commit to including a preliminary free-form generation result  (e.g., perplexity on a held-out corpus) in the appendix of the camera-ready version.
> ### Q2: What are the extra computational costs induced by the variational logit network for VGLR during training?
> During training, VGLR uses a single sample ($S=1$) via the reparameterisation trick, so the per-step training overhead is minimal:
> + **Forward pass:** One additional pass through the lightweight inference network ($D \rightarrow H \rightarrow N$, where $H = D/4$). This adds approximately 0.77% FLOPs for VGLR-MF and 1.07% for VGLR-FC (Table 4).
> + **Backward pass:** Gradients flow through the inference network $\phi$ only (the base model backbone and router weights $W_r$ are frozen in Stage 2). This is significantly cheaper than full backpropagation.
> + **Wall-clock time:** On A100 GPUs, Stage 2 training (10 epochs on OBQA) completes in 1–3 hours per dataset, which is modest relative to Stage 1 (LoRA fine-tuning, 3 epochs).
>
> We will add an explicit training time comparison table in the revision.

---

> > ### Author Rebuttal · Reviewer_EyAf · 2026-04-03
> >
> > I would like to thank the authors for the rebuttal. I'll keep my positive score.

---

### Official Review · Reviewer_sowo · 2026-03-13

**Soundness:** 3
**Presentation:** 3
**Significance:** 3
**Originality:** 3
**Overall Recommendation:** 5
**Confidence:** 3

**Summary:**

This paper is motivated by the brittleness of deterministic routing in MoE Transformers: small perturbations can cause unstable expert selection and poor calibration. After showing empirically that adding stochasticity to routing can be beneficial, this paper proposes a Bayesian variational routing framework with two variants: VGLR, which infers a posterior over routing logits, and VTSR, which uses an input-dependent temperature as a lighter approximation in selection space. Based on these settings, the following experiments on Granite-MoE, Qwen-MoE, and DeepSeek-MoE show that VGLR-FC yields large ECE improvements while preserving accuracy, and that both proposed methods improve routing stability over deterministic MAP routing.

**Compliance With Llm Reviewing Policy:**

Affirmed.

**Final Justification:**

This paper is motivated by the brittleness of deterministic routing in MoE Transformers: small perturbations can cause unstable expert selection and poor calibration. After the rebuttal, my concerns have been adequately addressed. The idea of further improving expert routing from a probabilistic perspective is interesting.

**Key Questions For Authors:**

- How should authors reconcile the low-temperature motivation with the high-entropy prior in VTSR?
- Why are there no comparisons with more recent MoE-specific Bayesian uncertainty methods?
- How much of the gain comes from Bayesian posterior modeling, and how much comes from simple stochastic routing?
- How important are the layer-selection heuristic and the sampling budget in practice?

**Limitations:**

yes

**Strengths And Weaknesses:**

**Strengths:**

- The move from a latent-variable view of routing to router-level variational inference is consistent, and the two proposed methods fit the stated goal of obtaining scalable uncertainty estimates without Bayesianizing the entire model.
- The baseline experiments are well designed. This paper covers several representative MoE architectures, including Qwen and DeepSeek series.
- In addition to calibration, the paper reports routing stability under perturbation, OOD detection behavior, and efficiency-related analysis.

**Weaknesses:**

- The motivating experiment suggests that lower-temperature routing can improve the accuracy–calibration trade-off. But VTSR is motivated through a uniform prior and entropy maximization, which pushes in the opposite conceptual direction.
- Recent work has already studied Bayesian uncertainty for expert pathways in Qwen1.5-MoE and DeepSeek-MoE, which is directly relevant here [1].

[1] Dialameh, Maryam, et al. "Bayesian Mixture of Experts For Large Language Models." *arXiv preprint arXiv:2511.08968* (2025).

---

> ### Author Rebuttal · Authors · 2026-03-30
>
> We thank Reviewer sowo for the thoughtful review. We address the key questions below and hope this strengthens their positive opinion of our work.
>
> ### Q1: How should authors reconcile the low-temperature motivation with the high-entropy prior in VTSR?
> This is an excellent question that highlights a subtlety of our framework. The low-temperature finding in Figure 1b demonstrates that *sharpening routing decisions improves calibration for in-distribution data*. The uniform (high-entropy) prior in VTSR does not contradict this — it serves as a **regulariser** that the model can override when the data likelihood strongly supports a confident decision. Further, in our motivating experiment the low-temperature is not input dependent and therefore does not capture an input-dependent uncertainty signal.
>
> Concretely, the ELBO objective (reconstruction − $\beta D_{\text{KL}}$) means the model learns $T_\phi(u)$ such that:
> + For **confident, in-distribution inputs**: the reconstruction term dominates, driving $T$ → low values (sharp decisions), consistent with Figure 1b.
> + For **ambiguous or out-of-distribution inputs**: the KL term pulls $T$ toward higher values (softer decisions), providing an appropriate uncertainty signal.
>
> This is precisely why VTSR's Gate-Ent signal excels at OoD detection (Table 2, AUROC 0.743): the learned temperature *amplifies* the entropy difference between ID and OoD inputs. We can provide histograms of learned temperature distributions on ID vs. OoD data in the revision to make this mechanism explicit.
>
> ### Q2: Why are there no comparisons with more recent MoE-specific Bayesian uncertainty methods (Dialameh et al., 2025)?
> Indeed the work of Dialameh et al. (2025) is cited within our original submission. This paper appeared on arXiv in November 2025, which provided us time to cite and conceptually compare in our original submission (citation on page 2) but we did not conduct an experimental validation in the weeks prior to submission. Their approach applies variational inference *within the expert networks* (weight-space), which is precisely the paradigm we critique in Section 2.2 as computationally indirect for routing uncertainty. VMoER is complementary: we target the routing decision variables directly, incurring < 1% overhead compared to the substantial cost of weight-space posteriors over expert parameters. We will add a further discussion of this concurrent work in the revision.
>
> ### Q3: How much gain comes from Bayesian posterior modelling vs. simple stochastic routing?
> This is a key ablation. The comparison between Temp-Scale (simple fixed stochasticity) and VTSR (learned, input-dependent stochasticity) directly isolates this effect. On Granite-MoE:
> + **Temp-Scale:** ECE = 0.107, ACC = 0.716 (improved calibration but *degraded* accuracy)
> + **VTSR:** ECE = 0.052, ACC = 0.736 (improved calibration without *accuracy* loss)
>
> The learned temperature adapts per-token, providing stochasticity where beneficial and determinism where needed — something a fixed temperature cannot achieve. Similarly, VGLR-FC's advantage over MCDropout (which also adds stochasticity but without a principled posterior) confirms that the Bayesian formulation contributes beyond mere noise injection.
>
> ### Q4: How important are the layer-selection heuristic and the sampling budget?
> The ablation in Appendix F (Table 10) shows that layer selection matters substantially. On Granite with VGLR: the Susceptible-10 strategy achieves ECE=0.015 vs. First-10 at ECE=0.242 — an order of magnitude difference. For sampling budget, we found diminishing returns beyond $S \approx 10–15$ samples (we use $S=35$ conservatively).

---

> > ### Author Rebuttal · Reviewer_sowo · 2026-04-02
> >
> > Overall, this is a well-executed manuscript of good quality. The idea of further improving expert routing from a probabilistic perspective is interesting. I will therefore raise my score.

---

### Official Review · Reviewer_XU7N · 2026-03-15

**Soundness:** 2
**Presentation:** 3
**Significance:** 2
**Originality:** 3
**Overall Recommendation:** 4
**Confidence:** 3

**Summary:**

The paper proposes to use amortized VI over the MoE router's logits for fine-tuning a MoE model on multiple-choice questions.
Evaluated on three models, the resulting model shows better calibration and OOD detection ability compared with models fine-tuned with point estimation.

**Compliance With Llm Reviewing Policy:**

Affirmed.

**Final Justification:**

I would like to thank the author for the detailed (final) response.

I am now slightly more convinced by the advantages of covariance modelling, but still not sure about the correctness for some of the arguments: e.g. "The limitations of the MF assumption in standard VAEs are well-documented; [1] introduced Normalising Flows to overcome this restriction." This is an **incorrect** interpretation of the normalization flow paper, their critism about the mean-field is for non-amortized VI, not for VAE. In fact, if I remember correctly, normalization flow also uses a factorized Gaussian as the prior.

> Forward Pass Requirements

I would appreciate it if the author could clarify this in the manuscript.

It is also worth noting that the authors frame this work as a "principled" framework, but the actual computation actually includes undocumented heuristics like this.

> Temperature Scaling and Accuracy

In my original review and the follow-up response, I am referring to temperature scaling on the output space not on the MoE router space. I apologize for the confusion. Given the topic of this submission and the keywords "Variational Inference, Bayesian Deep Learning, Uncertainty Quantification," I thought we were on the same page and thinking about the same temperature scaling.

## Summarizaiton

I've increased my score to 4.

Overall, I find the idea interesting and I think the idea is worth being published, but this is conditioned on the authors adjust the writing and the pitch in a more appropriate way: A lightweight Bayes fine-tuning approach tailored for MoE.

Many statements in the paper seem to be very concerning and give me the feeling that "the approach is better because it's Bayes", which is never true.

**Key Questions For Authors:**

- How does the approach compare with MAP + temperature scaling?

- How is gumbel softmax's temperature scheduled during training? Has the author considered using policy gradient rather than reparameterization trick?

**Limitations:**

Yes, limitations are discussed in the last paragraph of the conclusion section.

**Strengths And Weaknesses:**

# Strenghts

- The logit space VI is interesting in that the dimension would be much lower than directly applying VI over the weight space, which could be an advantage over methods that perform weight space VI (however, I am not aware of any, all the works I know perform VI over low-rank LoRA space)

- Due to the small dimension logit space, one can use ~~meanfield rather than fullrank~~ fullrank rather than meanfield for the variational posterior, increasing expressiveness, which is a questionable claim because the output of the residual mean head should already have correlation between dimensions implicitly captured by the NN's weights and explicit modeling of covariance may be unnecessary.

- I could see the value of the work as a more parameter-efficient approach for Bayesian fine-tuning (compared with e.g. Bayes LoRA / BLOB or LoRA ensemble), however, the authors do not demonstrate such evidences in the paper.

# Weaknesses

- My major criticism is that the work does not address practical MoE challenge, e.g. router imbalance, instability caused by router shift at post training time, etc. Instead, the paper studies fine-tuning MoE on very small multiple-choice dataset, where the model can easily overfit and become overconfident, and then the paper tries to address the calibration problem (which also happens to dense models), so this is a challenge from fine-tuning rather than from MoE.

- In the abstract, it should be made very clear that the paper only targets at fine-tuning tasks.

- The latent variable model is over-simplified. For example, the correlation between different models' expert selection is not considered.

- The proposed approach requires multiple forward propagations during inference time (to sample experts). This should be taken into account during evaluation. Different methods should be compared under same overhead.

## Minor

Since the authors propose to perform VI over temperature, I believe
`Mandt, Stephan, et al. "Variational tempering." Artificial intelligence and statistics. PMLR, 2016.`
should be cited.

---

> ### Author Rebuttal · Authors · 2026-03-30
>
> We thank reviewer XU7N for their feedback. We believe several points rest on misconceptions about our work; we hope the clarifications below will address these and that the reviewer will consider increasing their score.
>
> ### On the need for explicit covariance modelling
> The reviewer claims "the output of the residual mean head should already have correlation between dimensions implicitly captured." Within our framework, the residual mean head answers the question "for inputs like $x$ how much do I need to change my routing decision on average?" While it is true that this can capture some correlation between experts, when one is answering this question with the goal of providing uncertainty-aware information, this correction on its own is likely insufficient.
>
> To underscore the role and importance of covariance modelling, we highlight that the covariance answers a fundamentally different question to the residual head, namely: "for inputs like $x$ (heteroscedasticity), for which experts is my correction normally wrong, by how much, and what is the relationship of these errors?" This is a statement about the *shape* of uncertainty around the point estimate: a second-moment property that no first-moment correction (e.g., non-linear residual mean update), however expressive, can encode.
>
> ### Addressing Practical MoE Challenges
> We respectfully disagree with the assertion that this work does not tackle the core practical MoE challenges. VMoER does address several of the practical MoE challenges the reviewer mentions, both theoretically and empirically:
> + **Routing instability:** The reviewer claims we do not address "instability caused by router shift" as a core MoE challenge. However, Section 5.4 and Table 9 directly measure this: VMoER improves routing consistency under perturbation by up to 38% (Jaccard similarity 0.422 → 0.702 on Granite at σ=0.01). Importantly, the motivational brittleness analysis (Appendix B.1, Figure 9) is conducted on the pre-trained model without any fine-tuning, confirming that routing instability is a structural property of MoE architectures, not a fine-tuning artefact.
> + **Load imbalance as a prior:** Section 4.1 shows that existing load-balancing losses can be reinterpreted as KL penalties against a uniform prior, a theoretical contribution that unifies heuristic stabilisation strategies under a single Bayesian framework. This suggests that principled prior design could replace ad-hoc auxiliary losses in future work.
> + **Fine-tuning critique:** More broadly, we note that the reviewer seems to suggest that our contribution is solely addressing an over-fitting problem during fine-tuning that is shared with dense models. While overconfidence after fine-tuning is indeed general, the mechanism through which VMoER addresses it by modelling uncertainty over the discrete expert selection process is structurally specific to MoE. The high quality of the OoD uncertainty signal in practice (Table 2, 9-point AUROC improvement over standard entropy) demonstrates that our method captures meaningful routing uncertainty beyond simply correcting overconfidence and thus does more than just address overfitting.
>
> ### Forward-pass requirements
> The reviewer states as a criticism that our approach requires multiple forward passes through the model. This is incorrect: it does not require multiple forward passes through the model. We require multiple evaluations of the MoE routing only. For a complete discussion we point the reviewer to Section 5.5 of our paper and additionally point them to Appendix E where a detailed accounting and explanation is provided.
>
> ### Further questions
> The Temp-Scale baseline reported in Table 1 is precisely MAP + post-hoc temperature scaling with a globally optimised temperature. As shown, Temp-Scale improves calibration but its effect on accuracy is inconsistent. This instability stems from a single global scalar being unable to adapt to token-level difficulty. In contrast, VTSR learns an input-dependent temperature, achieving consistent calibration gains without accuracy degradation.
>
> We thank the reviewer for suggesting a joint latent variable model across all MoE layers and will discuss this as future work in the main text.
>
> We agree that in our abstract we can highlight that our experimental results are only for fine-tuning; however, we note that the theoretical framework remains general.
>
> The Gumbel-Softmax relaxation temperature is fixed at 1.0 throughout training. Gumbel-Softmax provides biased but low-variance gradients that proved more stable in practice.
>
> We further agree that `Mandt, Stephan, et al.` should be cited and we will commit to doing so in the camera-ready version of the paper.

---

> > ### Author Rebuttal · Reviewer_XU7N · 2026-04-01
> >
> > I would like to thank the authors for the response.
> >
> > Overall, I think the idea has great potential and is a promising way to incorporate more probabilistic inference into MoE.
> >
> > However, in my opinion, the current version needs more clarification on the current scope of the work: Fine-tuning MoE on small-scale datasets, where the model can easily show some fine-tuning specific issues, e.g. bad calibration / OOD robustness, that motivate the usage of Bayesian approaches.
> >
> > This is fine since this is the first paper along this line, as long as the authors clarify this and discuss the room for furtuer work, e.g. whether the approach can be useful in pre-training or continuous pre-training.
> >
> > Now for the authors' specific responses,
> >
> > ## covariance modelling
> >
> > I understand the motivation and I think it makes sense, but I think this only holds for linear models / non-amortized VI. In amortized VI I believe you can show the cholesky factor can be absorbed into the last layer in the encoder network, making the mean already correlated, that's why people don't use full rank gaussian as ~q(x | z)~ q(z | x) in VAE.
> >
> > My hypothesis for why introducing covariance head boosts the performance is that you have extra number of parameters in the network.
> >
> > ## Routing instability
> >
> > Thanks for the clarification, I am now convinced of this advantage.
> >
> > ## Load imbalance as a prior
> >
> > I agree that they share similiar principles and intuition, but it is unclear to me how the KL divergence translates to load balancing loss used in practice e.g. Eq 2 in https://arxiv.org/pdf/2501.11873
> >
> > I would also be careful about using words such as "principled", as you would still need some heuristics included, e.g. the coefficient in front of the KL term / exluding the KL term from Adam's normalizer or not.
> >
> > It's also very unclear to me what kind of new load-balancing approaches can come out of this Bayesian framework.
> >
> > ## Forward-pass requirements
> >
> > OK, after looking at sec 3.3 I think my confusion is resolved: The authors propose to sample multiple logits, do a local Bayeisan model averaging to get the softmax probability, then do a single layer-wise forward pass.
> >
> > But is this a proper unbiased estimation of, e.g. the marginal distribution described in Eq.4, or is it just a heuristic for alleviating noise?
> >
> > I would appreciate if the author could provide more explanation.
> >
> > ##  Temperature scaling
> >
> > I am confused by the authors' arguments, why would temperature scaling [1] affect the accuracy? Temperature scaling on the output space, i.e. over all posible options in the multiple choice QA datasets the author considered, should only adjust the magnitude of the logits but not adjusting the order.
> >
> > [1] https://arxiv.org/abs/1706.04599

---

> > > ### Author Response · Authors · 2026-04-07
> > >
> > > We thank reviewer XU7N for their thoughtful engagement with our work and we are pleased to have resolved many of their questions.
> > >
> > > We completely agree with the reviewer about clarifying the scope (highlighting our focus on fine-tuning) along with the addition of a subsection in the main text on the need for future works along the lines started by this paper. We commit to making these changes along with the other changes highlighted below.
> > > ### Covariance modelling
> > >
> > > We thank the reviewer for the follow-up as it helps us better-appreciate the confusion and we will address it in our exposition and appendix.
> > >
> > > The primary reason standard VAEs generally avoid full-rank Gaussian posteriors is not because the mean network correlates the latent dimensions, but due to the intractable cost of high-dimensional latent spaces. The limitations of the MF assumption in standard VAEs are well-documented; [1] introduced Normalising Flows to overcome this restriction.
> > >
> > > Recall our setting: the router output is a logit vector, $z$, that induces multinoulli parameters (via softmax), $p_z$. VMoER proposes a deterministic mean head that returns a vector $m$ and a covariance head which returns a matrix from which we can sample a vector $c$ i.e., $c \sim \mathcal{N}(0, \Sigma(u))$. We thus have final logits $z + m + c$. As both $z$ and $m$ are fully deterministic, we can only marginalize over the randomness of $c$ to induce our final multinoulli params $p_{z,m,c}$. NB: even with $m=0$ because the softmax is a non-linear transformation this is different from $p_z$.
> > >
> > >
> > > To the reviewers point, it is technically possible to learn a mean update $m^\*$ that is functionally equivalent i.e., $p_{z, m^\*} = p_{z, m, c}$. However, the technical existence of such a vector $m^\*$ is not conceptually insightful: (1) if one did learn such an input-dependent function returning $m^\*$ this is equivalent to the contribution of our VI procedure by definition (2) formulating our problem as being directly over the difficult-to-interpret space of $m^\*$ makes it unclear how one would express prior beliefs, perform inference, or disentangle uncertainties, and finally, (3) the decomposition of mean and covariance allows us to use $\Sigma(u)$, the posterior’s second moment, to capture uncertainty; which is valuable for downstream tasks (concrete results in Table 2). We will note that future works may perform inference over different latent spaces (e.g., akin $m^\*$) with various pros and cons.
> > >
> > > On the hypothesis of increased parameters: we hope that our explanation shows that the additional parameters in the covariance head derive their value from the conceptual role in Bayesian modelling and not from capacity alone.
> > >
> > >
> > > ### Load Imbalance as a prior
> > >
> > >
> > > We agree that “principled” should be weakened to e.g., “theoretically motivated.” Determining and testing the exact forms of load-balancing priors that may come out of our work is an important direction of future work and we commit to noting this as a part of our clarifications. Prospective future priors that may be promising include: more sophisticated data-driven priors learned from routing statistics rather than a fixed uniform prior, or exploring the possibility of new hierarchical priors that couple token- and network-level posteriors with batch-level balance constraints.
> > >
> > >
> > > ### Forward Pass Requirements
> > >
> > >
> > > We thank the reviewer for another perceptive question. Our local Bayesian Model Averaging provides an unbiased MC estimator of the routing probabilities, where $\bar{p} = \frac{1}{S} \sum_{s=1}^S \text{softmax}(z_s)$.
> > >
> > > However, for expert selection, it is a biased approximation as the Top-K operation is non-linear. However, this bias is a deliberate design choice. An unbiased estimator would evaluate the Top-K mask for every sample $s$, which could cause a single token to activate a large union of different experts across the $S$ samples which would eliminate the efficiency gains from sparsity. By taking the expected continuous probability before applying the Top-K routing, VMoER maintains strict K-expert sparsity while still allowing Bayesian uncertainty to dictate expert selection. In our revised manuscript we will make a clear note in the main text on this point.
> > >
> > >
> > > ### Temperature Scaling and Accuracy
> > >
> > >
> > > The accuracy degradation is due to the sampling procedure. We do not take the max of these probabilities but sample from them, thus the Top-K experts selected are not preserved for a given temperature. This is unlike what is done in Guo et al. which uses the argmax and thus keeps accuracy unchanged. We will ensure this is more clear in our revision.
> > >
> > >
> > > *Concluding remark:* We hope that the conceptual clarifications during the rebuttal process and our commitment to clarifying scope and future works will make the reviewer comfortable in increasing their score and recommending our paper for acceptance.
> > >
> > >
> > > [1] https://arxiv.org/pdf/1505.05770

---

### Decision · Program_Chairs · 2026-04-30

**Decision:**

Accept (regular)

**Comment:**

The paper reframes MoE routing as Bayesian inference in a practical, scalable way. The move to router-level variational inference is novel and the two proposed methods achieve uncertainty estimation at a computational overhead of only ~1%. The empirical evaluation is extensive, covering multiple architectures (Qwen, DeepSeek) across calibration, OOD detection, and robustness benchmarks, and the manuscript is generally well-written and easy to follow. The reviewers agree the work is worthy of publication. I would encourage the authors to follow Reviewer XU7N's request of rephrasing certain statements in the paper which seem to suggest the method is better simply by virtue of being Bayesian.